



# High Resolution Vertical Total Electron Content Maps Based on Multi-Scale B-spline Representations

Andreas Goss[1], Michael Schmidt[1], Eren Erdogan[1], Barbara Görres[2], and Florian Seitz[1]

[1]Deutsches Geodätisches Forschungsinstitut der Technischen Universität München, Arcisstraße 21, 80333 Munich
[2]Bundeswehr GeoInformation Centre (BGIC), Euskirchen, Germany

**Correspondence:** Andreas Goss (andreas.goss@tum.de)

**Abstract.** For more than two decades the IGS (International GNSS Service) Ionosphere Associated Analysis Centers (IAAC) provide global maps of the vertical total electron content (VTEC). In general, the representation of a two- or three-dimensional function can be performed by means of a series expansion or by using a discretization technique. Whereas in the latter case for a spherical function such as VTEC usually pixels or voxels are chosen, in case of a series expansion mostly spherical harmonics (SH) are used as basis functions. The selection of the best suited approach for ionosphere modelling means a trade-off between the distribution of available data and their possibility to represent ionospheric variations with high resolution and high accuracy.

Most of the IAACs generate Global Ionosphere Maps (GIMs) based on SH expansions up to the spectral degree $n = 15$ and provide them with a spatial resolution of $2.5° \times 5°$ with respect to latitude and longitude direction, and a temporal sampling of two hours. In the recent years it was frequently claimed to improve the spatial sampling of the VTEC GIMs to a spatial resolution of $1° \times 1°$ and to a temporal sampling of about 15 minutes. Enhancing the grid resolution means a interpolation of VTEC values for intermediate points but with no further information about variations in the signal. A degree 15 in the SH case for instance corresponds to a spatial sampling of $12° \times 12°$. Consequently, increasing the grid resolution requires at the same time an extension of the spectral content, i.e. to choose a higher SH degree value than 15.

Unlike most of the IAACs, the VTEC modelling approach at DGFI-TUM is based on localizing basis functions, namely tensor products of polynomial and trigonometric B-splines. This way, not only data gaps can be handled appropriately and sparse normal equation systems are established for the parameters estimation procedure, also a multi-scale-representation (MSR) can be set up, to determine GIMs of different spectral content directly by applying the so-called pyramid algorithm and to perform highly effective data compression techniques. The estimation of the MSR model parameters is finally performed by a Kalman-Filter driven by near real-time (NRT) GNSS data.

Within this paper we realize the MSR and create multi-scale products based on B-spline scaling and wavelet coefficients and VTEC grid values. We compare these products with different final and rapid products of the IAACs, e.g., the SH model from CODE (Berne) and the voxel solution from UPC (Barcelona). In opposite to that, DGFI-TUM's products are solely based on NRT GNSS observations and ultra-rapid orbits. Nevertheless, we can conclude that DGFI-TUMs high-resolution product ('othg') outperforms all products used within the selected time span of investigation, namely September 2017.



## 1 Introduction

The properties of the atmosphere can be described by means of different variables, e.g. the temperature or the charge state. In case of the temperature we distinguish with increasing height above the Earth's surface between troposphere (up to about 15 km height), stratosphere (about 15 to 50 km), mesosphere (about 50 to 90 km), thermosphere (about 90 to 800 km) and exosphere (above 800 km). In case of the charge state the atmosphere is split into the neutral atmosphere (up to 80 km height), the ionosphere (about 80 to 1000 km) and the plasmasphere (above 1000 km); see e.g. Limberger (2015).

The ionosphere is mostly driven by the Sun; extreme UV- (EUV-), X-ray and solar particle radiation cause ionization processes. In geodesy, the main ionospheric impact is the influence of free electrons on radio wave propagation. This effect mainly depends on the signal frequency, i.e., the ionosphere is a dispersive medium (Schaer, 1999). Signals with frequencies lower than 30 MHz will be blocked and reflected by the ionosphere, whereas signals with shorter wavelengths penetrate the ionosphere but are affected in speed and direction. The ionospheric influence on radio waves is twofold: the signal travel times are changed (delay) and the signal paths are modified (bending). Whereas the latter effect can be neglected for most applications, the ionospheric delay

$$d_{ion} = \pm \frac{40.3}{f^2} \int\limits_{S}^{R} N_e \, ds \tag{1}$$

depends directly on the electron density $N_e$ along the signal path $s$ between satellite $S$ and receiver $R$ and inversely on the carrier frequency $f$. Equation (1), which can be derived from dual-frequency measurements, is only an approximation since effects of higher order are neglected. These terms depend on signal frequency, signal elevation, and ionospheric conditions and reach about 0.2 cm in zenith for GPS signals (S. Bassiri, 1993). The sign on the right-hand side changes whether it is applied for a carrier phase observation ('−') or for a pseudorange measurement ('+'); see e.g. Langley (1998).

Observations of space geodetic techniques, such as the Global Navigation Satellite Systems (GNSS) and the Doppler Orbitography and Radiopositioning Integrated by Satellite (DORIS) tracking system as well as satellite altimetry and Ionospheric Radio Occultation (IRO) are based on electromagnetic signal propagation and thus, disturbed by the ionosphere. Most of the techniques are not directly sensitive for the electron density, but on the integrated effect along the ray path. In Eq. (1) the integral

$$STEC(\boldsymbol{x}^S, \boldsymbol{x}_R, t) = \int\limits_{S}^{R} N_e(\boldsymbol{x}, t) \, ds \tag{2}$$

is called slant total electron content (STEC). In Eq. (2) we introduce besides the time $t$ the position vectors $\boldsymbol{x}^S$, $\boldsymbol{x}_R$ and

$$\boldsymbol{x} = r \left[ \cos\varphi \cos\lambda, \ \cos\varphi \sin\lambda, \ \sin\varphi \right]^T. \tag{3}$$

of the satellite $S$, the receiver $R$ and an arbitrary point $P$ moving along the signal path $s$; the coordinate triple $(\varphi, \lambda, r)$ comprises latitude $\varphi$, longitude $\lambda$ and radial distance $r$ within a geocentric coordinate system $\Sigma_E$.



The vertical total electron content (VTEC)

$$VTEC(\varphi,\lambda,t) = \int\limits_{h_l}^{h_u} N_e(\varphi,\lambda,h,t)\,dh \tag{4}$$

is defined as the integration of the electron density in vertical direction, i.e. along the height $h$ above the Earth's surface, defined as $h = r - R_E$; $R_E$ means the radius of a spherical Earth. Furthermore in Eq. (4) $h_l$ and $h_u$ are the heights of the lower and

the upper boundary of the ionosphere; see e.g. Dettmering et al. (2011, 2014) and Limberger (2015).

The Eqs. (2) and (4) require a 3-D integration of the electron density. Often a simplification is preferred which is based on the so-called Single-Layer-Model (SLM). It assumes that all free electrons are concentrated in an infinitesimal thin shell, i.e. the sphere $\Omega_H$ with radius $R_H = R_E + H$ (Schaer, 1999) and $H$ being the single layer height. As a consequence of this assumption and according to

$$STEC(\boldsymbol{x}^S,\boldsymbol{x}_R,t) = M(z)\cdot VTEC(\boldsymbol{x}_{IPP},t) \tag{5}$$

VTEC can be transformed into STEC by introducing a mapping function $M(z)$ depending on the zenith angle $z$ of the ray path between satellite $S$ and receiver $R$. In Eq. (5) the position vector $\boldsymbol{x}_{IPP}$, i.e. the spherical coordinates $(\varphi_{IPP},\lambda_{IPP},R_H)$ define the Ionospheric Pierce Point (IPP), which means geometrically the piercing point of the straight line between $S$ and $R$ with the sphere $\Omega_H$ of the SLM. This point means the reference point of an observation including the STEC, such as a GNSS

measurement; see e.g. Erdogan et al. (2017). Figure 1 shows, for instance, the global distribution of the IPPs from GNSS observations at September 6, 2017 between 12:55 and 13:05 UT. However, it must be pointed out that the introduction of a simple isotropic mapping function $M(z)$, just depending on the zenith angle $z$, can only generate an approximation of STEC. Recently, more sophisticated approaches, e.g. the Barcelona Ionospheric Mapping Function (BIMF), have been developed to improve the projection of VTEC into STEC; see Lyu et al. (2018).

Combining the Eqs. (1), (2) and (5) yields the relation

$$d_{ion}(\boldsymbol{x}^S,\boldsymbol{x}_R,t) = -\frac{40.3}{f^2}\cdot M(z)\cdot VTEC(\boldsymbol{x}_{IPP},t) \tag{6}$$

between VTEC and the ionospheric delay $d_{ion}$ in case of a phase observation. Equation (6) can be interpreted and applied in two ways: if . . .

. . . VTEC is given from an ionospheric model, the delay $d_{ion}$ can be computed and used as a correction to GNSS observa-

tions,

. . . the delay $d_{ion}$ can be derived from double-frequency GNSS measurements, it can be used as an observation to develop or improve VTEC models.

Applications, such as satellite navigation and positioning require high precision and high resolution VTEC models. For that purpose the correction $d_{ion}$ could be according to Eq. (6) derived from VTEC maps, usually the so-called Global Ionosphere

Maps (GIM). The most prominent GIM is provided by the International GNSS Service (IGS) (Feltens and Schaer, 1998;





Hernández-Pajares et al., 2011) as a weighted combination product of VTEC maps from various IGS Ionosphere Associated Analysis Centers (IAAC), namely (1) the Jet Propulsion Laboratory (JPL), (2) the Center for Orbit Determination in Europe (CODE), (3) the European Space Operations Center of the European Space Agency (ESOC), (4) the Universitat Politècnica de Catalunya (UPC), (5) the Canadian Geodetic Survey of Natural Resources Canada (NRCan), (6) the Wuhan University (WHU)

and (7) the Chinese Academy of Sciences (CAS). Recently, Roma-Dollase et al. (2017) published a review paper on these seven GIMs concerning their mapping techniques and their consistency during one solar cycle.

There are several modeling strategies for generating GIMs; the most prominent approach is based on spherical harmonics (SH) and was introduced by Schaer (1999). Besides, the tomographic approach based on voxels (Hernández-Pajares et al., 1999) and other approaches based on B-spline scaling functions and wavelets (Schmidt, 2007; Schmidt et al., 2011; Schmidt M.,

2015), multivariate adaptive regression splines (MARS) and adaptive regression B-splines (BMARS) (Durmaz et al., 2010; Durmaz and Karslioglu, 2015) as well as polynomials (Komjathy and Langley, 1996) shall be mentioned here.

Generally, we distinguish between GIMs provided as *final*, *rapid*, *near real-time (NRT)* or *real-time (RT)* products. This classification is based on the latency of the underlying input data. In case of final products, for instance, only post-processed observations and orbits are used, NRT products are based on rapid orbits and observations with a latency of some minutes up

to a few hours. GIMs are typically provided with a temporal resolution of 2 hours or 1 hour and with a spatial resolution of $2.5° \times 5°$ with respect to geographical latitude and longitude, respectively (Hernández-Pajares et al., 2017).

VTEC variations are basically following annual, seasonal, diurnal and semi-diurnal periods. Earthquakes or incidental natural hazards can also cause small but visible signatures (Liu et al., 2004; Zhu et al., 2013). During space weather events however, such as a solar flares or a coronal mass ejections (CME), the number of free electrons may drastically increase. In the latter

case solar plasma consisting of electrons, ions and photons may enter the Earth's atmosphere and cause short period variations within the electron density distribution; see Monte-Moreno and Hernández-Pajares (2014); Wang et al. (2016); Tsurutani et al. (2006, 2009). As a consequence, the modeling of the disturbed ionosphere requires both a high temporal and a high spatial resolution. In 2012 during the IGS 2012 workshop in Olsztyn, Poland, it was recommended to provide high resolution IGS combined GIMs. The IAACs UPC and JPL agreed on disseminating GIMs with a temporal resolution of 15 minutes and a

spatial resolution of $1° \times 1°$ in latitude and longitude, respectively (Dach and Jean, 2013).

As already confirmed by Roma-Dollase et al. (2017), an increase in temporal resolution allows for an improvement in the overall accuracy of the GIMs. The authors compared the final products with a temporal resolution of 2 hours with rapid products with a temporal resolution of 15 minutes using the dSTEC analysis as the most reliable method to assess the accuracy of VTEC products (Hernández-Pajares et al., 2017). Following the results of their investigations, it can be stated that the increase of the

temporal resolution yields better results in the dSTEC analysis.

To the knowledge of the authors the spatial resolution of GIMs has not been investigated in detail, yet. Most of the GIMs are based on series expansions in terms of SHs with a maximum degree of $n_{max} = 15$. This value fits to a block size of about $12° \times 12°$ on the sphere $\Omega_H$. In opposite, a grid spacing of $2.5° \times 5°$ corresponds to a maximum SH degree of around $n = 36$; a $1° \times 1°$ grid spacing, i.e. a spatial resolution of around 110 km along the equator fits to a SH expansion up to degree $n = 180$.

As a matter of fact a reliable computation of the corresponding SH series coefficients requires a global input data coverage of





the same spatial sampling. Since the IAAC VTEC maps are solely based on GNSS observations with a rather inhomogeneous distribution (cf. Fig. 1 showing the IPPs of NRT observations with dense clusters over continents and large data gaps over oceans), finer ionospheric structures can only be monitored and modeled where high resolution input data are available. By

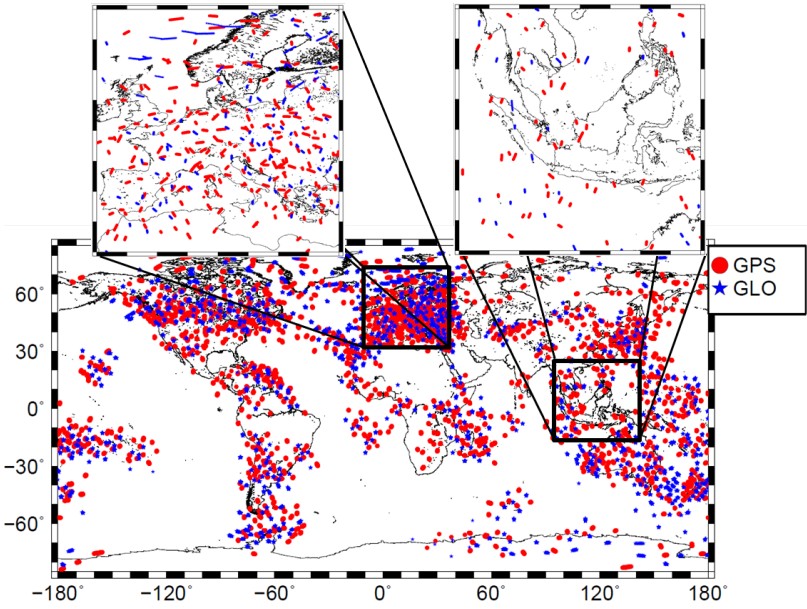

**Figure 1.** Global distribution of the IPPs from GPS (red dots) and GLONASS (blue stars) measurements for September 6, 2017 collected within a 10 minutes interval between 12:55 and 13:05 UT. The regional maps at the top are two 'zoom-ins' of Europe and Indonesia.

increasing the temporal resolution of the GIMs, the number of observations supporting the individual maps decreases. The two
zoom-in maps at the top of Fig. 1 show the strong incongruity between data distribution and signal structure. In areas with high resolution data, such as Europe, the U.S. or Australia, the VTEC signal is usually rather smooth (cf. the left panels in Fig. 9). In areas with high variable spatial and temporal signal structures such as in the equatorial belt, a much smaller number of observation is generally given. As a consequence, for global modeling we have to deal with a trade-off between signal structure and data resolution.

It is a well-known fact SHs as global basis functions are not suitable for representing unevenly globally distributed data. Consequently, in such a case, a series expansion in terms of localizing basis functions is more appropriate. In the sequel, we apply *tensor products* of *polynomial* and *trigonometric B-splines* as localizing 2-D basis functions. Besides the localizing features, B-splines additionally generate a multi-scale representation (MSR), also known as multi-resolution representation (MRR). The basis feature of a MSR is to split a target function into a smoothed, i.e., low-pass filtered version, and a number of
detail signals, i.e., band-pass filtered versions by successive low-pass filtering (Mertins, 1999). Hence, a spatial MSR of VTEC adapts the model resolution to the data distribution and thus, fulfills IGS' requirement of high resolution VTEC modeling.

    In this study, we compare global VTEC maps based on series expansions in terms of both globally defined SHs and localizing B-spline functions including the MSR with respect to the spectral content. For that purpose, we use the SH degree as the





common measure for the spectral content of a spherical signal. In detail we study the interrelations between the SH degree, the spatial sampling intervals of the input data and the resolution levels of B-spline expansions. In addition, we discuss the influence of different temporal resolutions of the GIMs. For the estimation of the unknown series coefficients of the B-spline expansion we use a Kalman Filter (KF) procedure as explained by Erdogan et al. (2017). In order to assess the quality of the

approach, we perform the dSTEC analysis (Hernández-Pajares et al., 2017).

The paper is outlined as follows: in Section 2 a description of VTEC modeling procedures based on both, SH and B-spline expansions are presented. In Subsection 2.3 we study the spectral resolution of global VTEC maps. Section 3 comprises a detailed description of the MSR and the estimation procedure. In Section 4 case studies are set up to verify the results of the previous sections numerically. Furthermore, this section provides a final assessment by means of the dSTEC validation. The

final section gives the conclusions and an outlook for future work.

## 2   VTEC Modeling Approaches

The 3-D signal $VTEC(\varphi, \lambda, t) = f(\boldsymbol{x}, t)$, introduced in the Eqs. (4) and (5), can be modeled as series expansion

$$f(\boldsymbol{x}, t) = \sum_{k=0}^{\infty} c_k(t)\, \phi_k(\boldsymbol{x}) \tag{7}$$

in terms of given space-dependent basis functions $\phi_k(\boldsymbol{x})$ and unknown time-dependent series coefficients $c_k(t)$. Assuming

that at discrete times $t_s = t_0 + s \cdot \Delta t$ with $s \in \mathbb{N}_0$ and sampling interval $\Delta t$ the altogether $I_s$ observations $y(\boldsymbol{x}_{i_s}, t_s)$ of VTEC given at IPP position $P_{i_s} \in \Omega_H$ with $i_s = 1, 2, \ldots, I_s$ are given. Considering the measurement errors $e(\boldsymbol{x}_{i_s}, t_s)$ the observation equation follows from Eq. (7) and reads

$$y(\boldsymbol{x}_{i_s}, t_s) + e(\boldsymbol{x}_{i_s}, t_s) = f_N(\boldsymbol{x}_{i_s}, t_s) =$$
$$= \sum_{k=0}^{N} c_k(t_s)\, \phi_k(\boldsymbol{x_{i_s}}) \,. \tag{8}$$

Note, that in the sequel of this paper we neglect the truncation error

$$r_N(\boldsymbol{x}_{i_s}, t_s) = \sum_{k=N+1}^{\infty} c_k(t_s)\, \phi_k(\boldsymbol{x_{i_s}}) \tag{9}$$

and omit other unknown parameters such as the satellite and receiver differential code biases (DCB) in case of GNSS geometry-free observations on the right-hand side of Eq. (8); see e.g. Erdogan et al. (2017).

In the following two Subsections 2.1 and 2.2 the SH expansion – as the probably most frequently used approach in ionosphere

modeling – and the 2-D B-spline tensor product approach are described.

---

[0]Note, for latitude $\varphi$ and longitude $\lambda$ we do not distinguish between geographical and geomagnetic spherical coordinates.



## 2.1 Spherical Harmonic Expansion

In the SH approach the observation equation (8) can be rewritten as

$$y(\boldsymbol{x}_{i_s}, t_s) + e(\boldsymbol{x}_{i_s}, t_s) = f_{n_{max}}(\boldsymbol{x}_{i_s}, t_s) =$$

$$= \sum_{n=0}^{n_{max}} \sum_{m=-n}^{n} c_{n,m}(t_s) Y_{n,m}(\boldsymbol{x}_{i_s}) \tag{10}$$

5 where the functions $Y_{n,m}(\boldsymbol{x})$, i.e. the SHs of degree $n = 0, \ldots, n_{max}$ and order $m = -n, \ldots, n$, are defined as

$$Y_{n,m}(\boldsymbol{x}) = P_{n,|m|}(\sin\varphi) \cdot \begin{cases} \cos m\lambda & \text{if} \quad m \geq 0 \\ \sin|m|\lambda & \text{if} \quad m < 0 \end{cases} \tag{11}$$

with $P_{n,|m|}$ being the normalized associated Legendre functions of degree $n$ and order $m$. The altogether $(n_{max}+1)^2$ quantities $c_{n,m}(t)$ in Eq. (10) are the time-dependent SH coefficients. According to the sampling theorem on the sphere the maximum degree $n_{max}$ is related to the sampling intervals $\Delta\varphi$ and $\Delta\lambda$ of the input data with respect to latitude $\varphi$ and longitude $\lambda$, namely

$$\Delta\varphi < \frac{180°}{n_{max}} \quad \text{and} \quad \Delta\lambda < \frac{180°}{n_{max}} . \tag{12}$$

As can be seen from Eq. (11) SHs are basis functions of global support. This implies that each single SH function is different from zero almost everywhere on the sphere $\Omega_H$. Consequently, each coefficient $c_{n,m}$ has to be recomputed, if only one additional observation is considered in the set of observation equations (10).

15 Since the VTEC observations $y(\boldsymbol{x}_{i_s}, t_s)$ at IPP positions will usually not be given on a spatial grid with constant mesh size, the sampling intervals $\Delta\varphi$ and $\Delta\lambda$ in the formulae (12) have to be interpreted as global averaga values.

## 2.2 B-Spline Expansion

At DGFI-TUM we rely on B-splines as basis functions for ionosphere modeling, since they are (1) characterized by their localizing feature and (2) they can be used to generate a MSR. For VTEC modeling we rewrite Eq. (8) as

20 $$y(\boldsymbol{x}_{i_s}, t_s) + e(\boldsymbol{x}_{i_s}, t_s) = f_{J_1,J_2}(\boldsymbol{x}_{i_s}, t_s) =$$

$$= \sum_{k_1=0}^{K_{J_1}-1} \sum_{k_2=0}^{K_{J_2}-1} d_{k_1,k_2}^{J_1,J_2}(t_s) \, \phi_{k_1,k_2}^{J_1,J_2}(\varphi_{i_s}, \lambda_{i_s}) \tag{13}$$

with initially unknown time-dependent scaling coefficients $d_{k_1,k_2}^{J_1,J_2}(t_s)$ and the 2-D scaling functions $\phi_{k_1,k_2}^{J_1,J_2}(\varphi_{i_s}, \lambda_{i_s})$ of levels $J_1$ and $J_2$ with respect to $\varphi$ and $\lambda$. The latter are defined as tensor products

$$\phi_{k_1,k_2}^{J_1,J_2}(\varphi, \lambda) = \phi_{k_1}^{J_1}(\varphi) \, \widetilde{\phi}_{k_2}^{J_2}(\lambda) \tag{14}$$





of 1-D scaling functions $\phi_{k_1}^{J_1}(\varphi)$ and $\widetilde{\phi}_{k_2}^{J_2}(\lambda)$ depending on latitude $\varphi$ and longitude $\lambda$, respectively. Since the B-spline approach is not as well known as the SH approach, it will be described in more detail in the following; we further refer to Dierckx (1984); Stollnitz et al. (1995a, b); Lyche and Schumaker (2001); Jekeli (2005); Schmidt M. (2015) and citations therein.

To decompose VTEC into its spectral components via the MSR in Section 3 the Eqs. (13) and (14) need to be rewritten in vector and matrix notation. For that purpose we introduce the $K_{J_1} \times 1$ vector

$$\boldsymbol{\phi}_{J_1}(\varphi) = \left[\, \phi_0^{J_1}(\varphi),\, \phi_1^{J_1}(\varphi),\, \ldots,\, \phi_{K_{J_1}-1}^{J_1}(\varphi)\,\right]^T ,\tag{15}$$

the $K_{J_2} \times 1$ vector

$$\widetilde{\boldsymbol{\phi}}_{J_2}(\lambda) = \left[\, \widetilde{\phi}_0^{J_2}(\lambda),\, \widetilde{\phi}_1^{J_2}(\lambda),\, \ldots,\, \widetilde{\phi}_{K_{J_2}-1}^{J_2}(\lambda)\,\right]^T\tag{16}$$

as well as the $K_{J_1} \times K_{J_2}$ coefficient matrix

$$\boldsymbol{D}_{J_1,J_2} = \begin{bmatrix} d_{0,0}^{J_1,J_2} & d_{0,1}^{J_1,J_2} & \cdots & d_{0,K_{J_2}-1}^{J_1,J_2} \\ d_{1,0}^{J_1,J_2} & d_{1,1}^{J_1,J_2} & \cdots & d_{1,K_{J_2}-1}^{J_1,J_2} \\ \vdots & \ddots & \ddots & \vdots \\ d_{K_{J_1}-1,0}^{J_1,J_2} & d_{K_{J_1}-1,1}^{J_1,J_2} & \cdots & d_{K_{J_1}-1,K_{J_2}-1}^{J_1,J_2} \end{bmatrix} .\tag{17}$$

Considering the computation rules for the Kronecker product '$\otimes$' (Koch, 1999) Eq. (13) can be written as

$$\begin{aligned} f(\varphi,\lambda,t) &= (\widetilde{\boldsymbol{\phi}}_{J_2}(\lambda) \otimes \boldsymbol{\phi}_{J_1}(\varphi))^T \,\mathrm{vec}\,\boldsymbol{D}_{J_1,J_2}(t) \\ &= \boldsymbol{\phi}_{J_1}^T(\varphi)\,\boldsymbol{D}_{J_1,J_2}(t)\,\widetilde{\boldsymbol{\phi}}_{J_2}(\lambda) \end{aligned}\tag{18}$$

wherein 'vec' means the vec-operator.

## 2.2.1 Polynomial B-splines

In the sequel we apply polynomial quadratic B-splines

$$\phi_{k_1}^{J_1}(\varphi) := N_{J_1,k_1}^2(\varphi)\tag{19}$$

of resolution level $J_1 \in \mathbb{N}_0$ and shift $k_1 = 0,1,...,K_{J_1}-1$ to represent the latitude dependent variations of VTEC. To be more specific, altogether $K_{J_1} = 2^{J_1}+2$ B-splines are located along a meridian depending on the latitude $\varphi \in [-90°,90°]$. To construct the $K_{J_1}$ B-spline functions the sequence

$$-90° = \varphi_0^{J_1} = \varphi_1^{J_1} = \varphi_2^{J_1} < \varphi_3^{J_1} < ... < \varphi_{K_{J_1}}^{J_1} =$$
$$= \varphi_{K_{J_1}+1}^{J_1} = \varphi_{K_{J_1}+2}^{J_1} = 90°\tag{20}$$

of knot points $\varphi_{k_1}^{J_1}$ is established; the consideration of multiple knot points at the poles is called 'endpoint-interpolating' and ensures the closing of the modeling interval. The constant distance between two consecutive knots $\varphi_{k_1}^{J_1}$ and $\varphi_{k_1+1}^{J_1}$ for





$k_1 = 2, ..., K_{J_1} - 1$ amounts $180°/2^{J_1}$. Following Schumaker and Traas (1991) and Stollnitz et al. (1995b) the normalized quadratic polynomial B-splines are calculable via the recursive relation

$$N^n_{J_1,k_1}(\varphi) = \frac{\varphi - \varphi^{J_1}_{k_1}}{\varphi^{J_1}_{k_1+n} - \varphi^{J_1}_{k_1}} N^{n-1}_{J_1,k_1}(\varphi)$$
$$+ \frac{\varphi^{J_1}_{k_1+n+1} - \varphi}{\varphi^{J_1}_{k_1+n+1} - \varphi^{J_1}_{k_1+1}} N^{n-1}_{J_1,k_1+1}(\varphi). \tag{21}$$

with $n = 1, 2$ from the initial values

$$N^0_{J_1,k_1}(\varphi) = \begin{cases} 1 & \text{if } \varphi^{J_1}_{k_1} \leq \varphi < \varphi^{J_1}_{k_1+1} \text{ and } \varphi^{J_1}_{k_1} < \varphi^{J_1}_{k_1+1} \\ 0 & \text{otherwise .} \end{cases}$$

Note, in Eq. (21) a factor is set to zero if the denominator is equal to zero.

As can be seen from Fig. 2, B-splines are characterized by their compact support or – in other words – they are different from zero only within a small subinterval of length $\Delta_{J_1} \approx 3 \cdot h_{J_1}$ where

$$h_{J_1} = \frac{180°}{2^{J_1} + 1} \tag{22}$$

means approximately the distance between two consecutive B-splines along the meridian. Since the total number $K_{J_1}$ of B-splines depends on the level $J_1$, finer structures can be modeled by increasing $J_1$. The numerical value for the level $J_1$ depends on the global average value $\Delta\varphi$ for the input data sampling interval in latitude direction according to

$$\Delta\varphi < h_{J_1} \tag{23}$$

(Schmidt et al., 2011). Solving Eq. (23) under the consideration of Eq. (22) for the level value $J_1$ the inequality

$$J_1 \leq \log_2\left(\frac{180°}{\Delta\varphi} - 1\right) \tag{24}$$

results.

### 2.2.2 Trigonometric B-splines

For modeling the longitudinal variations of VTEC trigonometric B-splines $T^3_{J_2,k_2}(\lambda)$ of order 3 and depending on the resolution level $J_2 \in \mathbb{N}_0$ and shift $k_2 = 0, 1, ..., K_{J_2} - 1$ are applied. As can be seen from Fig. 3 the altogether $K_{J_2} = 3 \cdot 2^{J_2}$ trigonometric B-splines are located along the parallels of the chosen spherical coordinate system within the interval $\lambda \in [0°, 360°)$. Consequently, the first and the last two B-spline functions within the interval $[0°, 360°)$ have to be completed by the so-called *wrapping around* effect. This constraint allows to define trigonometric B-splines in two different ways:

1. Following Schumaker and Traas (1991), Jekeli (2005) and Limberger (2015) periodic trigonometric B-splines can be calculated by means of a recurrence relation similar to Eq. (21). Thereby, additional constraints have to be introduced to force the periodicity of the series coefficients.



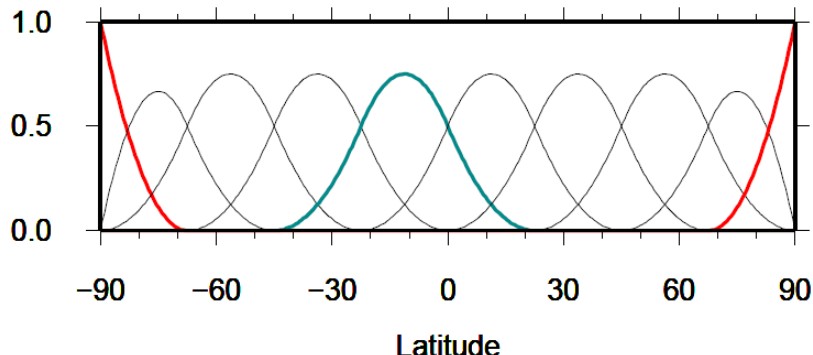

**Figure 2.** Polynomial B-splines of level $J_1 = 3$ with total number $K_{J_1} = 2^3 + 2 = 10$ of B-splines along the meridian. The *blue-colored* spline function $N^2_{3,4}(\varphi)$ corresponds to the shift value $k_1 = 4$ and covers a subinterval of length $\Delta_3 \approx 3 \cdot 180°/9 = 60°$. The *red-colored* spline functions $N^2_{3,0}(\varphi)$ with shift value $k_1 = 0$ and $N^2_{3,9}(\varphi)$ with shift value $k_1 = 9$ close the modeling interval at the poles.

2. The second option was introduced by Lyche and Schumaker (2001) and used by Schmidt et al. (2011); Schmidt M. (2015). It will be described in the following in more detail.

To be more specific, the sequence of non-decreasing knot points

$$0° = \lambda_0^{J_2} < \lambda_1^{J_2} < ... < \lambda_{k_2}^{J_2} < ... < \lambda_{K_{J_2}-1}^{J_2} < 360°, \tag{25}$$

with additional knots

$$\lambda_{K_{J_2}+i}^{J_2} = \lambda_i^{J_2} + 360° \quad \text{for} \quad i = 0, 1, 2 \tag{26}$$

for considering the periodicity is introduced. Similar as for the polynomial B-splines the distance between consecutive knots $\lambda_{k_2}^{J_2}$ and $\lambda_{k_2+1}^{J_2}$ for $k_2 = 0, 1, \ldots, K_{J_2}+1$ is given as

$$h_{J_2} = \frac{360°}{K_{J_2}} = \frac{120°}{2^{J_2}} \tag{27}$$

and thus, the length of the non-zero subinterval of a trigonometric B-spline function $T^3_{J_2,k_2}(\lambda)$ reads $\Delta_{J_2} = 3 \cdot h_{J_2} = 360°/2^{J_2}$. Following Lyche and Schumaker (2001) we define the functions

$$M_{J_2,k_2}(\lambda) = T^3_{J_2,k_2}(\lambda) = T^3_{h_{J_2}}(\lambda - \lambda_{k_2}^{J_2}). \tag{28}$$

Setting for simplification $h_{J_2} =: h$ and $\lambda - \lambda_{k_2}^{J_2} =: \Theta$, the functions $T^3_{h_{J_2}}(\lambda - \lambda_{k_2}^{J_2}) = T^3_h(\Theta)$ can be calculated via

$$T^3_h(\Theta) = \begin{cases} \frac{\sin^2(\Theta/2)}{\sin(h/2)\sin(h)} & \text{for} \quad 0 \le \Theta < h \\[2mm] \frac{1}{\cos(h/2)} - \frac{\sin^2((\Theta-h)/2)+\sin^2((2h-\Theta)/2)}{\sin(h/2)\sin(h)} \\[1mm] \qquad\qquad\qquad\qquad \text{for} \quad h \le \Theta < 2h \\[2mm] \frac{\sin^2((3h-\Theta)/2)}{\sin(h/2)\sin(h)} & \text{for} \quad 2h \le \Theta < 3h \\[2mm] 0 & \text{otherwise}. \end{cases} \tag{29}$$

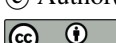


Finally, we define the basis functions

$$
\widetilde{\phi}_{k_2}^{J_2}(\lambda) = \begin{cases} M_{J_2,k_2}(\lambda) & \text{for } k_2 = 0, \dots, K_{J_2} - 3 \\[2mm] M_{J_2,k_2}(\lambda) + M_{J_2,k_2}(\lambda - 360^\circ) \\[2mm] \qquad\qquad \text{for } k_2 = K_{J_2} - 2, K_{J_2} - 1 \end{cases} \tag{30}
$$

introduced in Eq. (14). Figure 3 shows trigonometric B-splines of level $J_2 = 2$. With larger values for level $J_2$ the splines become more narrow and finer structures can be modeled. The choice of the level value $J_2$ again depends on the input data sampling interval. Analog to Eq. (23) the inequality

$$
\Delta\lambda < h_{J_2} \tag{31}
$$

has to be fulfilled where $\Delta\lambda$ denotes the global average value of the data sampling interval in longitude direction. Finally, under the consideration of Eq. (27) the inequality

$$
J_2 \leq \log_2\left(\frac{120^\circ}{\Delta\lambda}\right). \tag{32}
$$

for the level value $J_2$ is obtained.

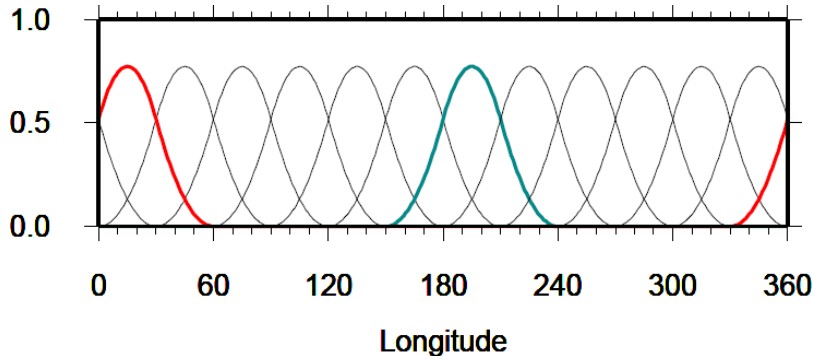

**Figure 3.** $K_{J_2} = 3 \cdot 2^2 = 12$ trigonometric B-splines $\widetilde{\phi}_{k_2}^{J_2}(\lambda)$, according to Eq. (30) for level $J_2 = 2$. The *blue-colored* spline function $\widetilde{\phi}_5^2(\lambda)$ with shift value $k_2 = 5$ is different from zero only in the subinterval of length $\Delta_{J_2} = 360^\circ/4 \approx 90^\circ$. The *red-colored* basis function $\widetilde{\phi}_{11}^2(\lambda)$ shows the *wrapping-around* effect.

### 2.3 Spectral Resolution of Global VTEC Models

In the Subsection 2.1 we derived the relations between the maximum degree $n_{max}$ of a SH expansion and the sampling intervals $\Delta\varphi$ and $\Delta\lambda$ of the input data. In the previous Subsection 2.2 the corresponding relations between the level values $J_1$ and $J_2$ of a B-spline expansion and the data sampling intervals have been deduced. The substitution of the expression $180^\circ/n_{max}$ from





the inequalities (12) into the Eqs. (24) and (32) yields the altogether six inequalities

$$J_1 \leq \log_2 \left( \frac{180°}{\Delta\varphi} - 1 \right) \leq \log_2 \left( n_{max} - 1 \right) ,$$

$$J_2 \leq \log_2 \left( \frac{120°}{\Delta\lambda} \right) \leq \log_2 \left( \frac{2 \cdot n_{max}}{3} \right) . \tag{33}$$

Given the numerical values 1 to 6 for the B-spline levels $J_1$ and $J_2$ Table 1 presents the corresponding largest numerical values for each, the SH degree $n_{max}$ as well as the sampling intervals $\Delta\varphi$ and $\Delta\lambda$ by evaluating the inequalities (33). From the

**Table 1.** Numerical values for the B-spline levels $J_1$ and $J_2$, the maximum SH degree $n_{max}$ and the input data sampling intervals $\Delta\varphi$ and $\Delta\lambda$ by evaluating the inequalities (33); the left part of the table presents the numbers along a meridian (upper inequalities in Eq. (33)), the right part the corresponding numbers along the equator and its parallels according to the lower inequalities in Eq. (33).

| | Latitude | | | | | | | Longitude | | | | | |
|---|---|---|---|---|---|---|---|---|---|---|---|---|---|
| $J_1$ | 1 | 2 | 3 | 4 | 5 | 6 | $J_2$ | 1 | 2 | 3 | 4 | 5 | 6 |
| $n_{max}$ | 3 | 5 | 9 | 17 | 33 | 63 | $n_{max}$ | 3 | 6 | 12 | 24 | 48 | 96 |
| $\Delta\varphi$ | 60 | 36 | 20 | 10.5 | 5.45 | 2.85 | $\Delta\lambda$ | 60 | 30 | 15 | 7.5 | 3.75 | 1.875 |

spectral point of view the six inequalities (33) comprise the following three scenarios:

1. If the global sampling intervals $\Delta\varphi$ and $\Delta\lambda$ are known, the mid parts of the inequalities (33) are given. The maximum degree $n_{max}$ is calculable from the right-hand side inequalities and may be inserted into the SH expansion (10). The

left-hand side inequalities yield the two level values $J_1$ and $J_2$ which can be inserted into the B-spline expansion (13).

2. With a specified numerical value for $n_{max}$ the right-hand parts of the inequalities (33) are given. The data input sampling intervals $\Delta\varphi$ and $\Delta\lambda$ can be determined from the mid parts of the inequalities. Next the two numerical values for the level values $J_1$ and $J_2$ are calculable from the left-hand side inequalities and can be inserted into the B-spline expansion (13).

3. If the processing time of VTEC maps has to be considered, the level values $J_1$ and $J_2$ are subject to certain restrictions, since as a matter of fact the number of numerical operations increases exponentially with the chosen numerical values for the levels. In this case, from the given left-hand side inequalities the data sampling intervals $\Delta\varphi$ and $\Delta\lambda$ can be determined from the mid parts. Finally, the right-hand side inequalities yield numerical values for the maximum SH degree $n_{max}$.

As already mentioned in the introduction most of the GIMs produced by the IAACs are based on series expansions in SHs up to a maximum degree of $n_{max} = 15$. Following the above listed second strategy and Table 1 we obtain for this example the approximations $J_1 = 4$ (for $n_{max} = 17$) and $J_2 = 3$ (for $n_{max} = 12$) for the two B-spline levels $J_1$ and $J_2$. Inserting these





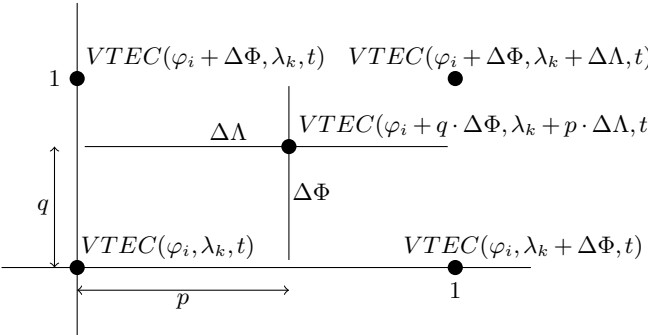

**Figure 4.** Schematic representation of the 4-point spatial interpolation to calculate the VTEC value at $P(\varphi_i + q \cdot \Delta\Phi, \lambda_k + p \cdot \Delta\Lambda)$ from the four corner points of the grid cell of interest.

numbers into the B-spline expansion (13) yields the spectarlly closest representation to the current IGS solutions. A numerical verification of this choice will be presented in Subsection 4.3.

## 2.4   VTEC Output Grids

The VTEC GIMs of the IAACs are usually provided with a spatial resolution of $\Delta\Phi = 2.5°$ in latitude direction and $\Delta\Lambda = 5°$
in longitude direction and a temporal sampling of $\Delta T = 2$ hours. Note, the resolution intervals $\Delta\Phi$, $\Delta\Lambda$ and $\Delta T$ are usually distinct from the sampling intervals $\Delta\varphi$, $\Delta\lambda$ and $\Delta t$ of the observations introduced in Subsection 2.1.

In order to calculate a VTEC value $VTEC(\varphi, \lambda, t)$ at an arbitrary location $P(\varphi = \varphi_i + q \cdot \Delta\Phi, \lambda = \lambda_k + p \cdot \Delta\Lambda)$ with $0 \leq q \leq 1$ and $0 \leq p \leq 1$ at an arbitrary time moment $t$ a simple bi-linear spatial interpolation from the VTEC values of the four given corner points $P(\varphi_i, \lambda_k)$, $P(\varphi_i, \lambda_k + \Delta\Lambda)$, $P(\varphi_i + \Delta\Phi, \lambda_k)$ and $P(\varphi_i + \Delta\Phi, \lambda_k + \Delta\Lambda)$ is performed according to

$VTEC(\varphi_i + q \cdot \Delta\Phi, \lambda_k + p \cdot \Delta\Lambda, t) =$

$$= (1 - q) \cdot (1 - p) \cdot VTEC(\varphi_i, \lambda_k, t)$$
$$+ \; q \cdot (1 - p) \cdot VTEC(\varphi_i + \Delta\Phi, \lambda_k, t)$$
$$+ \; p \cdot (1 - q) \cdot VTEC(\varphi_i, \lambda_k + \Delta\Lambda, t)$$
$$+ \; q \cdot p \cdot VTEC(\varphi_i + \Delta\Phi, \lambda_k + \Delta\Lambda, t) \, ; \tag{34}$$

see Schaer et al. (1998) and Fig. 4. Note, by applying the interpolation formula (34), the quality of the calculated VTEC value decreases with the spatial resolution intervals $\Delta\Phi$ and $\Delta\Lambda$ and depends on the position within the grid cell. In order to improve the quality of the VTEC computation two ways can be performed, namely

1.  the chosen model approach, e.g. the SH or the B-spline expansion can be used directly to calculate VTEC values at any arbitrary point $P(\varphi, \lambda)$,



2. the resolution intervals $\Delta\Phi$ and $\Delta\Lambda$ of the output grid can be set to smaller values, e.g., to $1°$ as it was proposed at the IGS workshop 2012.

For the calculation of a VTEC value $VTEC(\varphi, \lambda, t)$ at an arbitrary time moment $t = t_s + r \cdot \Delta T$ with $0 \leq r \leq 1$ at a given spatial location $P(\varphi, \lambda)$, an interpolation with respect to time can be applied. Commonly, the linear interpolation

$$VTEC(\varphi, \lambda, t) = (1-r) \cdot VTEC(\varphi, \lambda, t_s)$$
$$+ \ r \cdot VTEC(\varphi, \lambda, t_s + \Delta T) \tag{35}$$

between the two consecutive maps at epochs $t_s$ and $t_s + \Delta T$ is performed; see Schaer et al. (1998).

The previously described interpolation methods allow for the calculation of VTEC values $VTEC(\varphi, \lambda, t)$ at any spatial location $P(\varphi, \lambda)$ and at any time $t$. However, for a more accurate calculation of VTEC an increase of the resolution for both domains is necessary. Within the following section, it is shown that the usage of a MSR based on the B-spline approach in combination with a KF estimation procedure provides the possibility to create VTEC maps of higher spatial and temporal resolution. Consequently, according to Table 1 the calculated VTEC maps cover a wider spectral band, i.e. the numerical value of $n_{max}$ becomes larger.

## 3    Multi-Scale Representation

The B-spline functions as introduced in the Subsections 2.2.1 and 2.2.2 allow for the generation of a MSR. To be more specific, B-spline tensor product wavelet functions will be constructed which are intrinsically connected to the resolution levels of the MSR. Usually the MSR is interpreted as viewing on a signal under different resolutions such as a microscope does; see e.g. Schmidt (2012), Schmidt M. (2015); Schmidt et al. (2015) and Liang (2017). In all the aforementioned studies, the MSR is based on a regional 2-D representation of VTEC in terms of tensor products of polynomial B-spline functions only. Within this study, however, we apply the MSR for a global 2-D representation of VTEC in terms of tensor products of polynomial and trigonometric B-spline functions, as described by Lyche and Schumaker (2001) as well as Schumaker and Traas (1991).

### 3.1    Pyramid Algorithm

Neglecting the time dependency the B-spline approach (18) reads

$$f_{J_1, J_2}(\varphi, \lambda) = \boldsymbol{\phi}_{J_1}^T(\varphi) \, \boldsymbol{D}_{J_1, J_2} \, \widetilde{\boldsymbol{\phi}}_{J_2}(\lambda) . \tag{36}$$

In the context of the MSR the vectors $\boldsymbol{\phi}_{J_1}(\varphi)$ and $\widetilde{\boldsymbol{\phi}}_{J_2}(\lambda)$ are called scaling vectors, the elements $d_{k_1, k_2}^{J_1, J_2}$ of the matrix $\boldsymbol{D}_{J_1, J_2}$ are denoted as scaling coefficients.

With $J_1' = J_1 - J, J_2' = J_2 - J$ and $0 < J \leq \min(J_1, J_2)$ we obtain the 2-D MSR of the target function $f(\boldsymbol{x})$ introduced in Eq. (7) as

$$f_{J_1, J_2}(\varphi, \lambda) = f_{J_1', J_2'}(\varphi, \lambda) + \sum_{j=1}^{J} \sum_{\vartheta=1}^{3} g_{J_1-j, J_2-j}^{\vartheta}(\varphi, \lambda) . \tag{37}$$





Following the argumentation of Schmidt et al. (2015) but considering the polynomial and the trigonometric B-spline functions the low-passed filtered level-$(J_1', J_2')$ signal $f_{J_1', J_2'}(\varphi, \lambda)$ and the band-pass filtered level-$(J_1 - j, J_2 - j)$ detail signals $g^{\vartheta}_{J_1-j, J_2-j}(\varphi, \lambda)$ are computable via the relations

$$f_{J_1', J_2'}(\varphi, \lambda) = \boldsymbol{\phi}_{J_1'}^T(\varphi)\, \boldsymbol{D}_{J_1', J_2'}\, \widetilde{\boldsymbol{\phi}}_{J_2'}(\lambda)\,,$$

$$g^1_{j_1-1, j_2-1}(\varphi, \lambda) = \boldsymbol{\phi}_{j_1-1}^T(\varphi)\, \boldsymbol{C}^1_{j_1-1, j_2-1}\, \widetilde{\boldsymbol{\psi}}_{j_2-1}(\lambda)\,,$$

$$g^2_{j_1-1, j_2-1}(\varphi, \lambda) = \boldsymbol{\psi}_{j_1-1}^T(\varphi)\, \boldsymbol{C}^2_{j_1-1, j_2-1}\, \widetilde{\boldsymbol{\phi}}_{j_2-1}(\lambda)\,,$$

$$g^3_{j_1-1, j_2-1}(\varphi, \lambda) = \boldsymbol{\psi}_{j_1-1}^T(\varphi)\, \boldsymbol{C}^3_{j_1-1, j_2-1}\, \widetilde{\boldsymbol{\psi}}_{j_2-1}(\lambda) \tag{38}$$

where we introduced the definitions $j_1 = J_1 - j + 1$ and $j_2 = J_2 - j + 1$ for $j = 1, \ldots, J$. Herein, the $K_{j_1-1} \times 1$ and $K_{j_2-1} \times 1$ scaling vectors $\boldsymbol{\phi}_{j_1-1}(\varphi)$ and $\widetilde{\boldsymbol{\phi}}_{j_2-1}(\varphi)$ as well as the $L_{j_1-1} \times 1$ and $L_{j_2-1} \times 1$ wavelet vectors $\boldsymbol{\psi}_{j_1-1}(\varphi)$ and $\widetilde{\boldsymbol{\psi}}_{j_2-1}(\lambda)$ can

be calculated by means of the two-scale relations

$$\boldsymbol{\phi}_{j_1-1}^T(\varphi) = \boldsymbol{\phi}_{j_1}^T(\varphi)\, \boldsymbol{P}_{j_1}\,,$$

$$\widetilde{\boldsymbol{\phi}}_{j_2-1}^T(\lambda) = \widetilde{\boldsymbol{\phi}}_{j_2}^T(\lambda)\, \widetilde{\boldsymbol{P}}_{j_2}\,,$$

$$\boldsymbol{\psi}_{j_1-1}^T(\varphi) = \boldsymbol{\phi}_{j_1}^T(\varphi)\, \boldsymbol{Q}_{j_1}\,,$$

$$\widetilde{\boldsymbol{\psi}}_{j_2-1}^T(\lambda) = \widetilde{\boldsymbol{\phi}}_{j_2}^T(\lambda)\, \widetilde{\boldsymbol{Q}}_{j_2} \tag{39}$$

with $L_{j_1-1} = K_{j_1} - K_{j_1-1}$ and $L_{j_2-1} = K_{j_2} - K_{j_2-1}$.

The numerical entries of the $K_{j_1} \times K_{j_1-1}$ matrix $\boldsymbol{P}_{j_1}$ and the $K_{j_1} \times L_{j_1-1}$ matrix $\boldsymbol{Q}_{j_1}$ can be taken from Stollnitz et al. (1995b) or Zeilhofer (2008); the corresponding entries of the $K_{j_2} \times K_{j_2-1}$ matrix $\widetilde{\boldsymbol{P}}_{j_2}$ and the $K_{j_2} \times L_{j_2-1}$ matrix $\widetilde{\boldsymbol{Q}}_{j_2}$ are provided by Lyche and Schumaker (2001).

In the Eqs. (38) we introduced the $K_{j_1-1} \times K_{j_2-1}$ matrix $\boldsymbol{D}_{j_1-1, j_2-1}$ of scaling coefficients $d^{j_1-1, j_2-1}_{k_1, k_2}$ as well as the

$K_{j_1-1} \times L_{j_2-1}$ matrix $\boldsymbol{C}^1_{j_1-1, j_2-1}$, the $L_{j_1-1} \times K_{j_2-1}$ matrix $\boldsymbol{C}^2_{j_1-1, j_2-1}$ and the $L_{j_1-1} \times L_{j_2-1}$ matrix $\boldsymbol{C}^3_{j_1-1, j_2-1}$ of wavelet coefficients. These four matrices are calculable via the 2-D downsampling equation

$$\begin{bmatrix} \boldsymbol{D}_{j_1-1, j_2-1} & \boldsymbol{C}^1_{j_1-1, j_2-1} \\ \boldsymbol{C}^2_{j_1-1, j_2-1} & \boldsymbol{C}^3_{j_1-1, j_2-1} \end{bmatrix} = \begin{bmatrix} \bar{\boldsymbol{P}}_{j_1} \\ \bar{\boldsymbol{Q}}_{j_1} \end{bmatrix} \boldsymbol{D}_{j_1, j_2} \begin{bmatrix} \bar{\widetilde{\boldsymbol{P}}}_{j_2}^T & \bar{\widetilde{\boldsymbol{Q}}}_{j_2}^T \end{bmatrix} \tag{40}$$

also known as the 2-D pyramid algorithm. The $K_{j_1-1} \times K_{j_1}$ matrix $\bar{\boldsymbol{P}}_{j_1}$, the $K_{j_2-1} \times K_{j_2}$ matrix $\bar{\widetilde{\boldsymbol{P}}}_{j_2}$, the $L_{j_1-1} \times K_{j_1}$ matrix $\bar{\boldsymbol{Q}}_{j_1}$ and the $L_{j_2-1} \times K_{j_2}$ matrix $\bar{\widetilde{\boldsymbol{Q}}}_{j_2}$ are computable via the identities

$$\begin{bmatrix} \bar{\boldsymbol{P}}_{j_1} \\ \bar{\boldsymbol{Q}}_{j_1} \end{bmatrix} = \begin{bmatrix} \boldsymbol{P}_{j_1} & \boldsymbol{Q}_{j_1} \end{bmatrix}^{-1}\,, \tag{41}$$

$$\begin{bmatrix} \bar{\widetilde{\boldsymbol{P}}}_{j_2} \\ \bar{\widetilde{\boldsymbol{Q}}}_{j_2} \end{bmatrix} = \begin{bmatrix} \widetilde{\boldsymbol{P}}_{j_2} & \widetilde{\boldsymbol{Q}}_{j_2} \end{bmatrix}^{-1}\,; \tag{42}$$





see e.g. Schmidt (2007). The 2-D pyramid algorithm based on the decomposition (37) is visualized in Fig. 5. The '$0^{th}$' step transforms according to the Eqs. (13) and (18) the observations $y(\boldsymbol{x}_{i_j}, t_j)$ into the elements of the scaling matrix $\boldsymbol{D}_{J_1, J_2}(t_j)$ as introduced in Eq. (17). The applied procedure will be explained in Subsection 3.2.

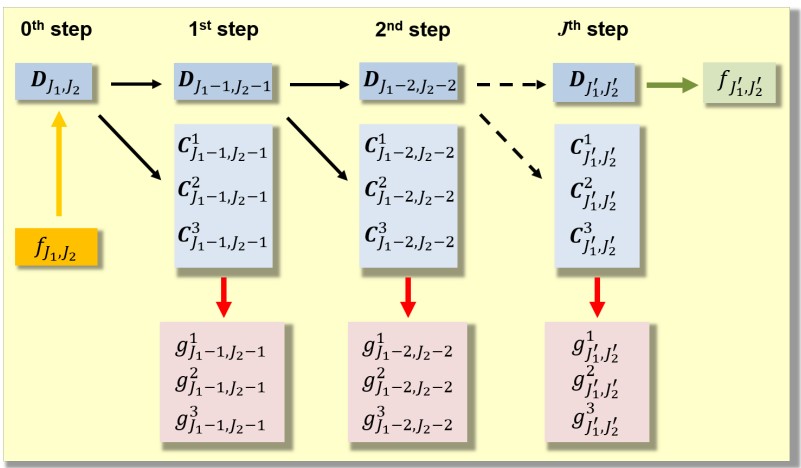

**Figure 5.** 2-D MSR of the signal $f_{J_1, J_2}(\varphi, \lambda)$.

The MSR as described before means a successive low-pass filtering of the target function $f(\varphi, \lambda, t)$ into two directions, namely latitude $\varphi$ and longitude $\lambda$, in the same manner. If a signal $f(\varphi, \lambda, t)$ is represented according to Eq. (18) up to the level values $J_1$ with respect to latitude and $J_2$ with respect to longitude, i.e. $f(\varphi, \lambda, t) \approx f_{J_1, J_2}(\varphi, \lambda, t)$ the application of the MSR (37) allows for the computation of low-pass filtered signal approximations up to the level pairs $(J_1 - 1, J_2 - 1), (J_1 - 2, J_2 - 2), \ldots$. The principal structures of the ionospheric key parameters such as VTEC, however, are usually parallel to the geomagnetic equator. Consequently, we will additionally deal with a 1-D MSR of the signal $f(\varphi, \lambda, t)$ with respect to the latitude. In this case Eq. (37) reduces to

$$f_{J_1, J_2}(\varphi, \lambda) = f_{J_1', J_2}(\varphi, \lambda) + \sum_{j=1}^{J} g_{J_1 - j, J_2}(\varphi, \lambda) . \tag{43}$$

Thus, signal approximations up to the level pairs $(J_1 - 1, J_2), (J_1 - 2, J_2), \ldots$ are obtained. From the four relations in Eqs. (38) only the first and the third one have to be considered within the 1-D MSR (43), namely

$$f_{J_1', J_2}(\varphi, \lambda) = \boldsymbol{\phi}_{J_1'}^T(\varphi) \, \boldsymbol{D}_{J_1', J_2} \, \widetilde{\boldsymbol{\phi}}_{J_2}(\lambda) ,$$
$$g_{j_1 - 1, J_2}(\varphi, \lambda) = \boldsymbol{\psi}_{j_1 - 1}^T(\varphi) \, \boldsymbol{C}_{j_1 - 1, J_2}^2 \, \widetilde{\boldsymbol{\phi}}_{J_2}(\lambda) \tag{44}$$

with $j_1 = J_1 - j + 1$ for $j = 1, \ldots, J$, $0 < J \le J_1$ and $J_1' = J_1 - J$. The $K_{j_1 - 1} \times K_{J_2}$ matrix $\boldsymbol{D}_{j_1 - 1, J_2}$ of scaling coefficients and the $L_{j_1 - 1} \times K_{J_2}$ matrix $\boldsymbol{C}_{j_1 - 1, J_2}^2$ of wavelet coefficients are calculable from the 1-D downsampling equation

$$\begin{bmatrix} \boldsymbol{D}_{j_1 - 1, J_2} \\ \boldsymbol{C}_{j_1 - 1, J_2}^2 \end{bmatrix} = \begin{bmatrix} \bar{\boldsymbol{P}}_{j_1} \\ \bar{\boldsymbol{Q}}_{j_1} \end{bmatrix} \boldsymbol{D}_{j_1, J_2} \tag{45}$$



where the matrices $\bar{\boldsymbol{P}}_{j_1}$ and $\bar{\boldsymbol{Q}}_{j_1}$ are computable via Eq. (41). The 1-D pyramid algorithm based on the decomposition (43) is visualized in Fig. 6.

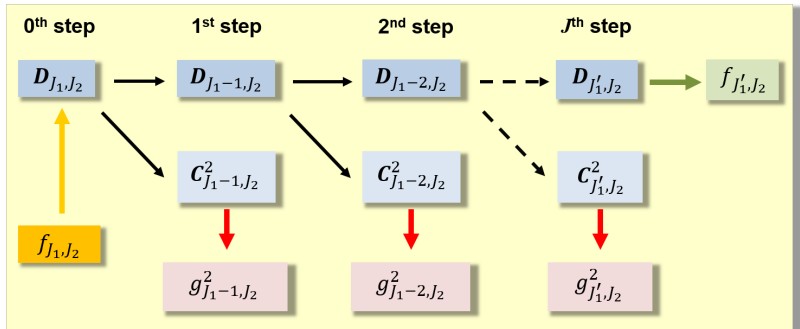

**Figure 6.** 1-D MSR of the signal $f_{J_1,J_2}(\varphi,\lambda)$ with respect to the latitude $\varphi$. The '0$^{\text{th}}$' step on the left-hand side conforms with the one of Fig. 5.

Besides the representation of a signal, e.g. VTEC, by means of approximations on different resolution levels with respect to latitude and longitude, the MSR also allows for the utilization of a powerful *data compression procedure*, since in general the numerical value of a large number of wavelet coefficients is close to zero depending on the signal structure; see e.g. Zeilhofer (2008).

## 3.2  Estimation of Unknown Model Parameters

To estimate the elements of the unknown $K_{J_1} \times K_{J_2}$ matrix $\boldsymbol{D}_{J_1,J_2}(t_s)$ from VTEC observations $y(\boldsymbol{x}_{i_s}, t_s)$ within the '0$^{\text{th}}$' step of the MSR we apply Kalman filtering according to Erdogan et al. (2017).

In the linear formulation the Kalman Filter consists (1) of the state equation

$$\boldsymbol{\beta}_s = \boldsymbol{F}_s\,\boldsymbol{\beta}_{s-1} + \boldsymbol{w}_{s-1} \tag{46}$$

and (2) of the system

$$\boldsymbol{y}_s + \boldsymbol{e}_s = \boldsymbol{A}_s\,\boldsymbol{\beta}_s \tag{47}$$

of observation equations. In Eq. (46 ) the $u \times 1$ vector $\boldsymbol{\beta}_s = \text{vec}\,\boldsymbol{D}_{J_1,J_2}(t_s)$ – known as the state vector – of the $u = K_{J_1} \cdot K_{J_2}$ unknown scaling coefficients at time moment $t_s$ is predicted from the state vector $\boldsymbol{\beta}_{s-1}$ of the previous time moment $t_{s-1}$ by means of the $u \times u$ transition matrix $\boldsymbol{F}_s$ and the $u \times 1$ vector $\boldsymbol{w}_{s-1}$ of the process noise. In Eq. (47) $\boldsymbol{y}_s = (y(\boldsymbol{x}_{i_s}, t_s))$ and $\boldsymbol{e}_s = (e(\boldsymbol{x}_{i_s}, t_s))$ are the $I_s \times 1$ vectors of the observations and the measurement errors, respectively; the $(i_s)^{\text{th}}$ row vector $\boldsymbol{a}_{i_s}^T$ of the $I_s \times u$ coefficient matrix $\boldsymbol{A}_s$ is given by the expression

$$\boldsymbol{a}_{i_s} = \widetilde{\boldsymbol{\phi}}_{J_2}(\lambda_{i_s}) \otimes \boldsymbol{\phi}_{J_1}(\varphi_{i_s}) \tag{48}$$



as introduced in Eq. (18). The measurement error vector $\boldsymbol{e}_s$ and the vector $\boldsymbol{w}_s$ of the process noise are assumed to be white noise vectors with expectation values $E(\boldsymbol{e}_s) = \boldsymbol{0}$ and $E(\boldsymbol{w}_s) = \boldsymbol{0}$, and fulfill the requirements

$$E(\boldsymbol{w}_s\,\boldsymbol{w}_l^T) = \boldsymbol{\Sigma}_{\boldsymbol{w}}\,\delta_{s,l}\,,\ E(\boldsymbol{e}_s\,\boldsymbol{e}_l^T) = \boldsymbol{\Sigma}_y\,\delta_{s,l}\,,\ E(\boldsymbol{w}_s\,\boldsymbol{e}_l^T) = \boldsymbol{0} \qquad (49)$$

where $\delta_{s,l}$ is the delta symbol which equals to 1 for $s = l$ and to 0 for $s \neq l$. In the Eqs. (49) $\boldsymbol{\Sigma}_y$ and $\boldsymbol{\Sigma}_{\boldsymbol{w}}$ are given covariance

matrices of the observations and the process noise, respectively.

The solution of the estimation problem as defined in the Eqs. (46) and (47) consists generally of the sequential application of a prediction step (time update) and a correction step (measurement update). In the prediction step, the estimated state vector $\widehat{\boldsymbol{\beta}}_{s-1}$ and its covariance matrix $\widehat{D}(\widehat{\boldsymbol{\beta}}_{s-1}) = \widehat{\boldsymbol{\Sigma}}_{\beta,s-1}$ are propagated from the time moment $t_{s-1}$ to the next time moment $t_s$ by means of

$$\boldsymbol{\beta}_s^- = \boldsymbol{F}_s\,\widehat{\boldsymbol{\beta}}_{s-1}\,, \qquad (50)$$
$$\boldsymbol{\Sigma}_{\beta,s}^- = \boldsymbol{F}_s\,\widehat{\boldsymbol{\Sigma}}_{\beta,s-1}\boldsymbol{F}_s^T + \boldsymbol{\Sigma}_w \qquad (51)$$

where the symbol '-' indicates the predicted quantities. The prediction step is followed by the measurement update

$$\widehat{\boldsymbol{\beta}}_s = \boldsymbol{\beta}_s^- + \boldsymbol{K}_s\,(\boldsymbol{y}_s - \boldsymbol{A}_s\,\boldsymbol{\beta}_s^-)\,, \qquad (52)$$
$$\widehat{\boldsymbol{\Sigma}}_{\beta,s} = (\boldsymbol{I} - \boldsymbol{K}_s\,\boldsymbol{A}_s)\,\boldsymbol{\Sigma}_{\beta,s}^- \qquad (53)$$

where $\widehat{\boldsymbol{\beta}}_s$ and $\widehat{\boldsymbol{\Sigma}}_{\beta,s}$ are the updated state vector and its covariance matrix. In the Eqs. (52 ) and (53) the $u \times I_s$ Kalman gain matrix

$$\boldsymbol{K}_s = \boldsymbol{\Sigma}_{\beta,s}^-\,\boldsymbol{A}_s^T\,(\boldsymbol{A}_s\,\boldsymbol{\Sigma}_{\beta,s}^-\,\boldsymbol{A}_s^T + \boldsymbol{\Sigma}_y)^{-1} \qquad (54)$$

behaves as a weighting factor between the new measurements and the predicted state vector. The chosen step size $t_s - t_{s-1}$ within the KF determines the maximum temporal resolution of the output.

Using the estimations $\widehat{\boldsymbol{\beta}}_s$ and $\widehat{\boldsymbol{\Sigma}}_{\beta,s}$ from the Eqs. (52) and (53), a $V \times 1$ vector $\boldsymbol{f}_s$ of function values $f(\varphi_i, \lambda_k, t_s)$ at arbitrary locations $P(\varphi_i, \lambda_k)$ with $i = 1, \ldots, I$, $k = 1, \ldots, K$ and $V = I \cdot K$ can be estimated by

$$\widehat{\boldsymbol{f}}_s = \bar{\boldsymbol{A}}_s\,\widehat{\boldsymbol{\beta}}_s\,, \qquad (55)$$
$$\widehat{\boldsymbol{\Sigma}}_{f,s} = \bar{\boldsymbol{A}}_s^T\,\widehat{\boldsymbol{\Sigma}}_{\beta,s}\,\bar{\boldsymbol{A}}_s \qquad (56)$$

where $\widehat{\boldsymbol{\Sigma}}_{f,s}$ is the estimated $V \times V$ covariance matrix of the estimation $\widehat{\boldsymbol{f}}_s$. The $V \times u$ matrix $\bar{\boldsymbol{A}}_s$ is set up in a similar way as

the matrix $\boldsymbol{A}_s$ in Eq. (47) with (48). In the following we will interpret the function values $f(\varphi_i, \lambda_k, t_s) = VTEC(\varphi_i, \lambda_k, t_s)$ as VTEC values.

### 3.3  B-Spline Model Output

The procedure explained before allows for the dissemination of two products, namely





– **Product 1**:

set of estimated scaling coefficients

$$\widehat{d}_{k_1,k_2}^{J_1,J_2}(t_s)\big|_{k_1=0,...,K_{J_1}-1,k_2=0,...,K_{J_2}-1} \tag{57}$$

from Eq. (52) at time moments $t_s$ for level values $J_1$ and $J_2$ at the spatial positions $k_1$ in latitude direction and $k_2$ in longitude direction, respectively, as well as their estimated standard deviations

$$\widehat{\sigma}_{d;k_1,k_2}^{J_1,J_2}(t_s)\big|_{k_1=0,...,K_{J_1}-1,k_2=0,...,K_{J_2}-1} \tag{58}$$

extracted from the covariance matrix (53) and

– **Product 2**:

estimated VTEC values

$$\widehat{VTEC}_{J_1,J_2}(\varphi_i,\lambda_k,t_s)\big|_{i=1,...,I,k=1,...,K} \tag{59}$$

according to Eq. (55) at time moments $t_s$ for level values $J_1$ and $J_2$ in latitude and longitude direction, respectively, calculated at grid points $P(\varphi_i,\lambda_k)$ as well as their estimated standard deviations

$$\widehat{\sigma}_{VTEC}^{J_1,J_2}(\varphi_i,\lambda_k,t_s)\big|_{i=1,...,I,k=1,...,K}\,, \tag{60}$$

extracted from the covariance matrix (56)

to the user. According to Eq. (35) the time interval $\Delta T$ between two consecutive maps of the coefficients (57) and their standard deviations (58) or the VTEC grid values (59) and their standard deviations (60) at times $t_s$ and $t_s + \Delta T$ can be chosen arbitrarily, e.g. as 10 or 15 minutes, 1 hour or 2 hours. Following Eq. (34) the coordinates $\varphi_i$ and $\lambda_k$ of the altogether $V$ grid points $P(\varphi_i,\lambda_k)$ are defined as $\varphi_i = -90° + (i-1) \cdot \Delta\Phi$ with $\Delta\Phi = 180°/(I-1)$ and $\lambda_k = 0° + (k-1) \cdot \Delta\Lambda$ with $\Delta\Lambda = 360°/K$. As mentioned before the spatial resolution intervals $\Delta\Phi$ and $\Delta\Lambda$ are usually chosen as $1°, 2.5°$ or $5°$, i.e. $I = 181, 73, 37$ and $K = 360, 144, 72$.

The two products, i.e. the set of coefficients or the VTEC grid values reflect the two strategies of dissemination. In case of a SH expansion for RT applications as introduced in Subsection 2.1 the corresponding SH coefficients $c_{n,m}$ from Eq. (10) can be transferred to the user by means of a RTCM (Radio Technical Commission for Maritime services) standard 1264 message. This message allows the consideration of SH coefficients, but only up to degree $n = 16$. In case of the B-spline expansion (13), however, an encoder procedure for the B-spline coefficients (57) is necessary, because the user has to evaluate the B-spline model just as in the SH case by substituting the B-spline tensor product (14) for the SHs (11). Due to the two restrictions, namely to use SH expansions only and just up to a maximum degree $n_{max} = 16$, the RTCM message format for data dissemination has to be discussed urgently and must be set up in a more flexible way, cf. the comments in Section 5. To apply the RTCM format in its current form, the VTEC grid values (59) can alternatively be used as observations $y(\boldsymbol{x}_{i_s}, t_s)$ in Eq. (10) to calculate SH coefficients $c_{n,m}(t_s)$ by means of a least-squares estimation. This way each GIM can be sent at a high update rate to the user for RT applications.





In case of Product 2 the VTEC grid values (59) as well as there standard deviations (60) are disseminated as VTEC and standard deviation maps, i.e. GIMs, with given spatial resolutions $\Delta\Phi$ and $\Delta\Lambda$ in latitude and longitude direction, respectively, in IONEX format to the user.

## 4  Numerical Investigations

5    In the sequel the described modeling approach developed at DGFI-TUM is applied to real data. To be more specific, we use GPS and GLONASS NRT data in hourly blocks and apply ultra-rapid orbits. A detailed explanation of the data pre-processing and the set up of the full observation equations is presented by Erdogan et al. (2017). The IGS IAACs provide final products based on post-processed GNSS observations and orbits with a latency of more than one week. Several IAACs provide in addition rapid products with a latency of one day by using rapid orbits. An overview on the products used in the sequel of this paper is given in Table 2.

**Table 2.** List of GIM products used in this paper. Information on names, types and latencies are taken from the references [1]: Roma-Dollase et al. (2017), [2]: Orus et al. (2005) and [3]: this paper.

| Institution | Product | Type | Latency | Reference |
|---|---|---|---|---|
| CODE | codg | final | > 1 week | [1] |
| UPC | uqrg | rapid | > 1 day | [2] |
| DGFI-TUM | oplg, ophg | NRT | < 3 hours | [3] |

For the evaluation of the data we have to define an appropriate coordinate system. Here we follow the standard procedure and use a Sun-fixed geomagnetic coordinate system. To be more specific, we identify the coordinate system $\Sigma_E$ introduced in the context of Eq. (3) with the Geocentric Solar Magnetic (GSM) coordinate system; see Laundal and Richmond (2017). Consequently, the SH and B-spline theory as presented in the previous sections is applied in the orthogonal GSM system. Since

15    in this coordinate system diurnal variations of the ionosphere are mitigated, the transition matrix $\boldsymbol{F}_2$ introduced in the state equation (46 ) of the KF can be set to the identity matrix $\boldsymbol{I}$, i.e. $\boldsymbol{F}_s = \mathbf{I}$. In other words, the dynamic system of the KF is set to a random walk process. Furthermore, for the time update in Eq. (46) we fix the step size $t_s - t_{s-1}$ to 5 minutes.

Whereas the scaling coefficients (57) and their standard deviations (58) of Product 1 are located within the GSM system, the VTEC values (59) and their standard deviations (60) of Product 2 are provided in the aforementioned IONEX format on a

20    regular grid defined in a geographical geocentric Earth-fixed coordinate system. Thus, a coordinate system transformation has to be interposed.



## 4.1 Validation procedure

For validation purposes we rely on the *dSTEC analysis* which is currently regarded as the standard method for the quality assessment of VTEC models; see e.g. Orus et al. (2007) and Rovira-Garcia et al. (2015).

This analysis method is based on the calculation of the difference between STEC observations $STEC(\boldsymbol{x}^S, \boldsymbol{x}_R, t_s)$ at discrete time moments $t_s$ according to Eq. (2) and a reference observation $STEC(\boldsymbol{x}^S, \boldsymbol{x}_R, t_{ref})$ along a specified satellite arc as

$$
\begin{aligned}
dSTEC_{obs}(t_s) &= STEC(\boldsymbol{x}^S, \boldsymbol{x}_R, t_s) \\
&\quad - STEC(\boldsymbol{x}^S, \boldsymbol{x}_R, t_{ref}) \, .
\end{aligned}
\tag{61}
$$

The reference time moment $t = t_{ref}$ is usually referred to the observation with the smallest zenith angle $z = z_{ref}$. In the same manner, the differences

$$
\begin{aligned}
dSTEC_{map}(t_s) &= M(z_s) \cdot VTEC(\boldsymbol{x}_{IPP}, t_s) \\
&\quad - M(z_{ref}) \cdot VTEC(\boldsymbol{x}_{IPP, t_{ref}})
\end{aligned}
\tag{62}
$$

are calculated by means of Eq. (5) from the VTEC map to be validated. The quality assessment is performed by studying the differences

$$
dSTEC(t_s) = dSTEC_{obs}(t_s) - dSTEC_{map}(t_s)
\tag{63}
$$

with expectation value $E(dSTEC(t_s)) = 0$.

## 4.2 Estimation of B-spline Multi-Scale Products

Figure 7 shows the global distribution of the IPPs related to GNSS VTEC observations $y(\boldsymbol{x}_{IPP}, t_s) = VTEC(\boldsymbol{x}_{IPP}, t_s)$ as introduced in Eq. (13) for September 6, 2017 at 13:00 UT. Since the B-spline model is set up in the GSM coordinate system and the scaling coefficients are restricted to the state equation

$$
d_{k_1,k_2}^{J_1,J_2}(t_s) = d_{k_1,k_2}^{J_1,J_2}(t_{s-1}) + w(t_{s-1})
\tag{64}
$$

according to Eq. (46), we select $\Delta\varphi = 5°$ and $\Delta\lambda = 10°$ as appropriate values for the global average sampling interval of the input data as introduced at the end of Subsection 2.1. Consequently, the B-spline levels to $J_1 = 5$ and $J_2 = 3$ are taken from Table 1.

The covariance matrices $\boldsymbol{\Sigma}_y$ and $\boldsymbol{\Sigma}_w$ of the observations and the process noise, respectively, as defined in the formulae (49), are set up according to Erdogan et al. (2017). In more detail, $\boldsymbol{\Sigma}_y$ consists of two diagonal block matrices related to GPS and GLONASS VTEC observations. The relative weighting between the blocks, i.e. between GPS and GLONASS, is performed by manually defined variance factors.

The top left panel of Fig. 8 shows with $J_1 = 5, J_2 = 3, K_{J_1} = 2^{J_1} + 2 = 34$ and $K_{J_2} = 3 \cdot 2^{J_2} = 24$ the numerical values of the total $816 = 34 \cdot 24$ scaling coefficients

$$
\widehat{d}_{k_1,k_2}^{5,3}(t_s)\big|_{k_1=0,\dots,33,\, k_2=0,\dots,23}
\tag{65}
$$





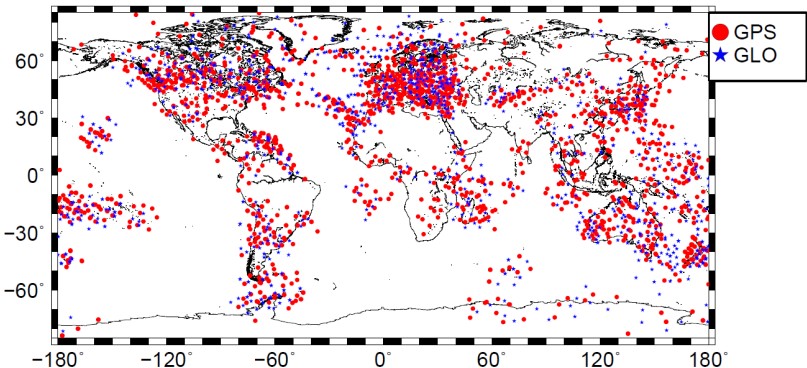

**Figure 7.** Global distribution of the IPPs from GPS (red dots) and GLONASS (blue stars) measurements for September 6, 2017, at 13:00 UT.

according to Eq. (57), estimated by means of Eq. (52). Since the shift values $k_1$ and $k_2$ determine the location of the scaling coefficients they can be plotted. The top right panel shows the corresponding standard deviations as defined in Eq. (58). A test of significance is performed to each of the scaling coefficients according to Koch (1999).

Whereas the two upper panels show the results of Product 1 in the GSM system, the two lower panels of Fig. 8 depict the corresponding results of Product 2 in a geographical geocentric coordinate system. With the choices $\Delta\Phi = 2.5°$ and $\Delta\Lambda = 5.0°$ for the grid spacing in latitude and longitude direction, respectively, Product 2 provides the VTEC grid values

$$\widehat{VTEC}_{5,3}(\varphi_i, \lambda_k, t_s)\big|_{i=1,\dots,73,k=1,\dots,72} \tag{66}$$

and the corresponding standard deviations $\widehat{\sigma}_{VTEC}^{5,3}$ from the Eqs. (59) and (60). Note, for the visualization of VTEC and their standard deviations in the two lower panels of Fig. 8 we computed functions values on a much denser grid by using the interpolation formula (34).

From the comparison of the two left-hand side panels in Fig. 8 it can be stated that the numerical values of the scaling coefficients directly reflect the signal structure, i.e. the signal energy. This fact is the consequence of the localizing character of the B-spline functions. The two right-hand side panels reveal that in general the standard deviations are larger where no or only a few GNSS observations to IGS stations are available, namely over the oceans, e.g. the Southern Atlantic, but also over specific continental regions such as the Sahara and the Amazon region.

The top left panel, i.e. the plot of the set (65) of the scaling coefficients $\widehat{d}_{k_1,k_2}^{5,3}$ can be interpreted as a visualization of the $34 \times 24$ matrix $\boldsymbol{D}_{5,3}$ defined in Eq. (17) and typed in the top left box of the Figs. 5 and 6 for the 2-D and the 1-D MSR. Consequently, the two upper panels of Fig. 8 are the results of the $0^{th}$ step within the pyramid algorithm as explained in Subsection 3. Applying the $1^{st}$ step of the 1-D pyramid algorithm the downsampling equation (44) provides both the $18 \times 24$ matrix $\boldsymbol{D}_{4,3}$ of estimated scaling coefficients $\widehat{d}_{k_1,k_2}^{4,3}$ for the level values $J_1 = 4$ and $J_2 = 3$ as well as the $16 \times 24$ matrix $\boldsymbol{C}_{4,3}^2$ of estimated wavelet coefficients. Consequently, the definition of the Product 2 in Subsection 3.3 can be extended to



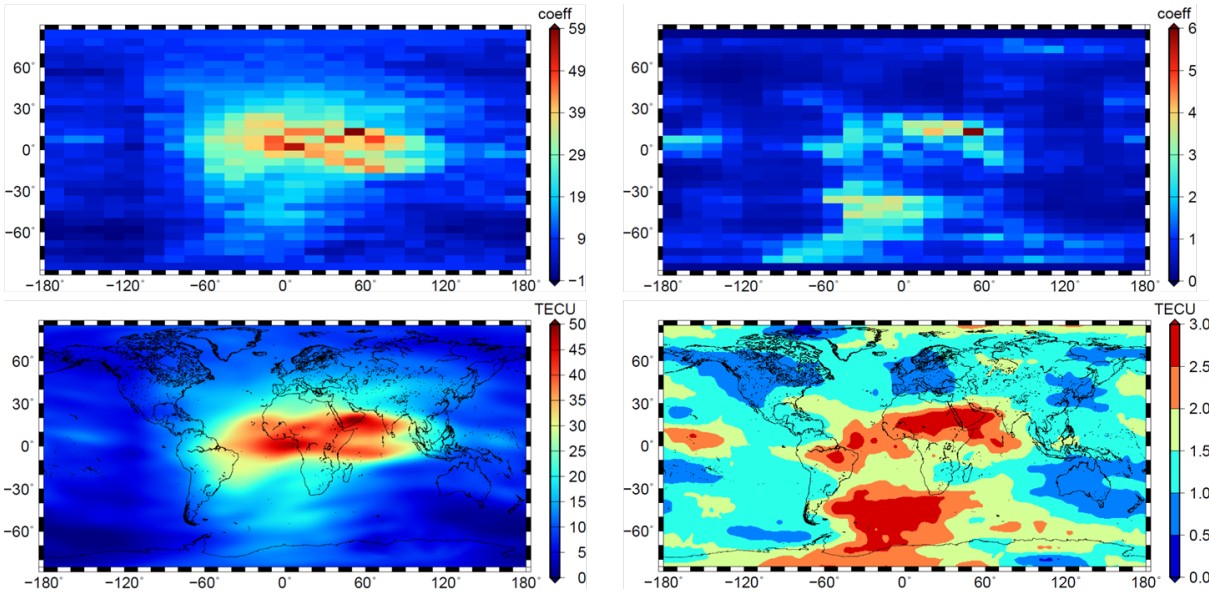

**Figure 8.** Estimated scaling coefficients (top left) and their standard deviations (top right) for level values $J_1 = 5$ and $J_2 = 3$ within the GSM coordinate system. Estimated VTEC values (bottom left) and their standard deviations (bottom right) as GIMs within a geographical coordinate system; all sets calculated for September 6, 2017 at 13:00 UT.

– **Multi-Scale Products 2**

**ophg**: estimations with levels $J_1 = 5$, $J_2 = 3$

$$\widehat{VTEC}_{5,3}(\varphi_i, \lambda_k, t_s) \, , \quad \widehat{\sigma}_{VTEC}^{4,3}(\varphi_i, \lambda_k, t_s) \tag{67}$$
$$\Delta\Phi = 2.5° \, , \quad \Delta\Lambda = 5.0°$$

5      **oplg**: estimations with levels $J_1 = 4$, $J_2 = 3$

$$\widehat{VTEC}_{4,3}(\varphi_i, \lambda_k, t_s) \, , \quad \widehat{\sigma}_{VTEC}^{4,3}(\varphi_i, \lambda_k, t_s) \tag{68}$$
$$\widehat{g}_{4,3}(\varphi_i, \lambda_k, t_s) \, , \quad \widehat{\sigma}_g^{\,4,3}(\varphi_i, \lambda_k, t_s)$$
$$\Delta\Phi = 2.5° \, , \quad \Delta\Lambda = 5.0°$$

We denote the two Multi-Scale Products 2 as 'ophg' and 'oplg', where the first digit is referred to the processing software
10   OPTIMAP which was developed within a third-party funded project (see Acknowledgements). The second digit 'p' is chosen
according to the time interval $\Delta T$ of map generation, i.e. 't' for $\Delta T = 10$ minutes, '1' for $\Delta T = 1$ hour and '2' for $\Delta T = 2$
hours. The third digit describes the spectral resolution and is selected as 'l' for 'low' and 'h' for 'high', finally, the last digit
indicates the model domain and is set to 'g' for 'global'. Furthermore, we want to mention again, that the products 'ophg' and
'oplg' are all presented in geographical coordinates.





### 4.3 Comparison of VTEC maps from B-spline and spherical harmonic expansions

As mentioned in the context of Table 1 the B-spline levels $J_1 = 4$ for latitude and $J_2 = 3$ for longitude fit the best to the highest degree $n_{max} = 15$ of a SH expansion (10). To be more specific, we compare the Multi-Scale Product 'o1lg' with the product 'codg' provided by CODE. 'codg' is characterized by a SH expansion up to degree $n_{max} = 15$ and a time interval $\Delta T = 1$
hour of two consecutive maps (Schaer, 1999).

Figure 9 shows the VTEC and standard deviation maps for September 6, 2017 at 13:00 UT as well as the difference map between 'o1lg' and 'codg'. Although the structures of the two VTEC maps are rather similar, the difference map shows deviations of up to $\pm6$ TECU. To judge this amount a comparison of VTEC GIMs from different IAACs was performed (not shown here). This investigation stated that deviations between individual IAAC products are also in the range of up to $\pm10$ %
or even more. Studying the structures within the difference map no larger systematic patterns are visible and, thus, justify our assumption that the quality of 'o1lg' is comparable with the quality of the IAAC products. The standard deviation maps on the right-hand side of Fig. 9 show different structures which are mainly caused by the application of the different estimation strategies, namely KF ('o1lg') and least-squares estimation ('codg'). To assess the comparability numerically we apply the dSTEC analysis described in Subsection 4.1. First we define a network of receiver stations which are used in Eq. (61). The
chosen set should not be used within the computation of the VTEC maps. Fulfilling both requirements at the same time is difficult and, thus, the set of stations shown in Fig. 10 contains both independent stations and stations used simultaneously in all VTEC models. Since GNSS measurements are taken along the satellite arcs, the corresponding IPPs are located spatially within a grid cell and temporally between the discrete time moments of the products 'o1lg' and 'codg'. In order to calculate the VTEC values in Eq. (62), the spatial and temporal interpolation formulae (34) and (35) have to be applied. Figure 11 shows the
RMS values of the differences (63) during the time span between September 1 and September 30, 2017, at the chosen receiver stations. As it can be seen, the RMS values vary between 0.3 and 1.6 TECU. By comparing the RMS values of 'o1lg' with a mean RMS value of 0.80 TECU and 'codg' with a mean RMS of 0.77 TECU we can state that the quality of these two products is very close to each other.

The results indicate that the overall quality of the NRT product 'o1lg' is comparable with that of the final product 'codg'
including the developed and implemented pre-processing strategies and steps of the GNSS data; cf. Table 2.

### 4.4 Assessment of the Multi-Scale VTEC Products

The two Multi-Scale VTEC Products 'ophg' and 'oplg' have been introduced in the two equation blocks (67) and (68). In what follows, we study them during a solar storm of medium intensity at September 8, 2017 and during the strongest storm of the last 10 years, the prominent St. Patrick storm, happening at March 17, 2015. Figure 12 shows on the left-hand side the results
of the September 8, 2017 event at 19:00 UT and at the right-hand side the corresponding maps for the St. Patrick storm event at March 17, 2015 at 19:00 UT.

As already mentioned in the context of Eq. (43) it is expected that the detail signal (44) is dominated by structures parallel to the geomagnetic equator. The detail signal $g_{4,3}(\varphi_i, \lambda_k, t_s)$ shown in the two bottom panels meets these expectations. Especially





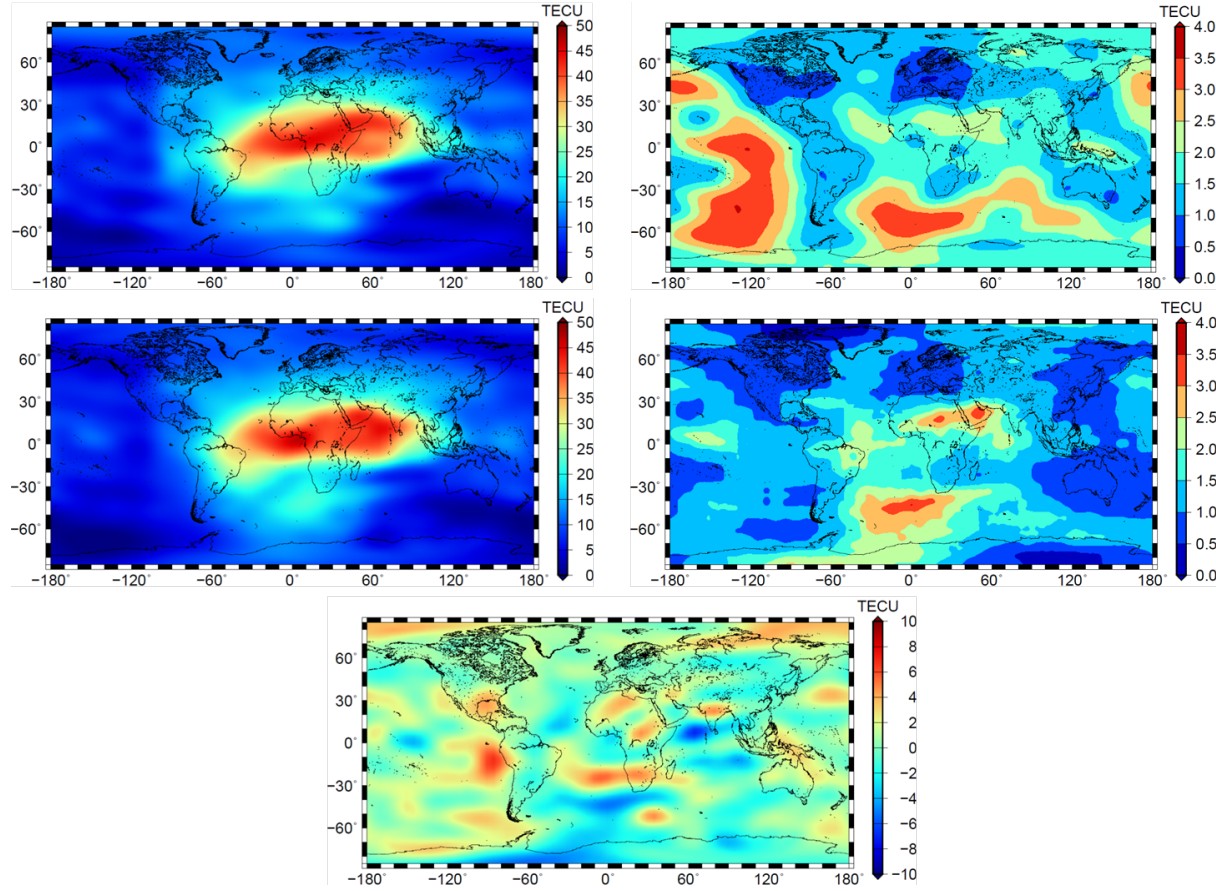

**Figure 9.** VTEC maps 'codg' (top left panel) and 'o1lg' (mid left panel) as well as their standard deviation maps (right-hand side panels); difference map of the two VTEC maps (bottom panel); all data for September 6, 2017 at 13:00 UT.

during the St. Patrick storm event the detail signal shows strong signatures. It should be mentioned that a large number of estimated wavelet coefficients collected in the matrix $C_{4,3}^2$ are characterized by absolute values smaller than a given threshold. The neglect of these coefficients allows for a high data compression rate. Consequently, the number of significant coefficients as the outcome of a MSR would go drastically below the number of scaling coefficients within the set (57) of Product 1; the reader can get an impression on the number of neglected coefficients by paying attention to the light green and light blue colors in the panels of the two detail signals in Fig. 12.

Next we focus on the solar storm during September 2017 and study the sampling intervals of different GIMs. As already mentioned, for the second digit in 'ophg' and 'oplg' the set $p \in \{t, 1, 2\}$ was defined for the sampling intervals $\Delta T = 10$ minutes, $\Delta T = 1$ hour and $\Delta T = 2$ hours of the VTEC models. Therefore the products 'othg' and 'otlg' are related to the sampling interval $\Delta T = 10$ minutes, the products 'o1hg' and 'o1lg' to $\Delta T = 1$ hour and the products 'o2hg' and 'o2lg' to $\Delta T = 2$ hours. In summary, we distinguish between six products of different spectral content and different sampling intervals.




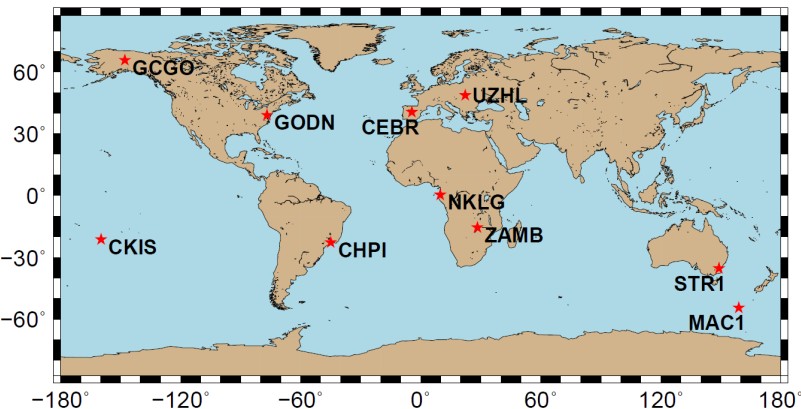

**Figure 10.** Distribution of the 10 IGS receiver stations used for the dSTEC analysis.

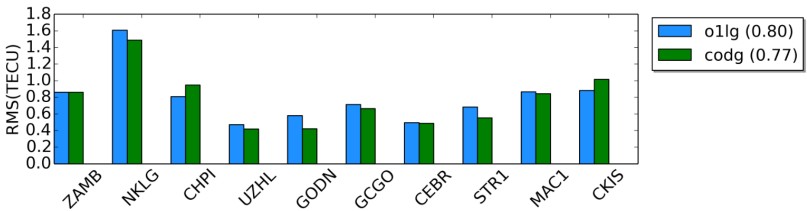

**Figure 11.** RMS values for the products 'codg' (green) and 'o1lg' (blue) computed at the 9 receiver stations shown in Fig. 10. The values in the parentheses in the legend are the average RMS values over all 10 receiver stations for the entire test period between September 1 and September 30, 2017.

Figure 13 depicts the RMS values computed by the dSTEC analysis at the stations shown in Fig. 10. It is assumed that a product with a larger sampling interval $\Delta T$ is less accurate than a product with a smaller sampling interval. Consequently, the average RMS values of 'o2hg' and 'o2lg' are larger compared to the corresponding values for a shorter sampling interval. Furthermore, it is assumed that RMS values for a product of higher B-spline levels, e.g. 'othg', are smaller than for the corresponding product

5   of lower B-spline level values such as 'otlg'. By comparing the corresponding color bars in Fig. 13, i.e. orange ('o2lg') vs. red ('o2hg'), light blue ('o1lg') vs. blue ('o1hg') and green ('otlg') vs. yellow ('othg'), the aforementioned assumptions are confirmed.

The differences of the RMS values of the first three products, 'o2lg', 'o1lg' and 'otlg', are caused by their different sampling intervals. Comparing the mean RMS values of 0.92 TECU and 0.80 TECU for 'o2lg' and 'o1lg', respectively, we find a

10   relative improvement of approximately 13.0 %. Decreasing the sampling from $\Delta T = 2$ hours to $\Delta T = 10$ minutes a further improvement of 16.3 % can be achieved. Comparing the RMS values 0.90 TECU, 0.72 TECU and 0.68 TECU of the products 'o2hg', 'o1hg' and 'othg', respectively, we find relative improvements of 20 % and 24.4 % by downsizing the sampling interval from 2 hours to 1 hour and finally to 10 minutes. A summary of the relative improvements is given in Table 3.





**Figure 12.** Multi-Scale VTEC Products for solar storm events: high resolution VTEC map 'ophg' for September 8, 2017 (top left panel) and for March 17, 2015 (top right panel); low resolution VTEC map 'oplg' for September 8, 2017 (mid left panel) and for March 17, 2015 (mid right panel); the bottom panels show the detail signals introduced in the equation block (68) and computed by means of Eq. (44) for the two solar events.

In the next step we compare the quality of the Multi-Scale Products 'ophg' and 'oplg' directly. First, we compare 'o2lg' with 'o2hg' and obtain an improvement of approximately 2.2 % . In the same manner for comparisons of 'o1lg' with 'o1hg' and 'otlg' with 'othg' improvements of 10.0 % and 11.7 % can be obtained. Table 4 shows the results for the comparison of each pair of products; an improvement is in indicated by green colored numbers, a worsening by blue colored numbers. As a consequence, an increase of the numerical value for level $J_1$, i.e. the enhancement of the spectral resolution with respect to the latitude yields a significant improvement in the RMS values as long as the temporal sampling $\Delta T$ is less than 2 hours. From the investigations in Subsection 4.3 it could be concluded that the quality of product 'o2lg' is comparable with the quality of the IAAC products. Further investigations have been done but are not presented here. It can be seen from Table 4 that there is







**Figure 13.** RMS values for the products 'o2hg', 'o1hg", 'othg', 'o2lg', 'o1lg' and 'otlg' computed at the 9 receiver stations shown in Fig. 10 during September 2017. The values in the parentheses in the legend are the average RMS values over all 10 receiver stations for the entire test period between September 1 and September 30, 2017.

**Table 3.** Relative improvements in percentage for a downsizing of the sampling interval of the products 'o2lg', 'o1lg', 'otlg', 'o2hg', 'o1hg' and 'otlg'.

| Product | RMS [TECU] | Percentage | Improvement |
|---------|-----------|------------|-------------|
| o2lg | 0.92 | 100% | |
| o1lg | 0.80 | 87.0% | 13.0% |
| otlg | 0.77 | 83.7% | 16.3% |
| o2hg | 0.90 | 100% | |
| o1hg | 0.72 | 80.0% | 20.0% |
| othg | 0.68 | 75.6% | 24.4% |

a strong improvement of more than 26 % by using the product 'othg' instead of 'o2lg'. It is worth to be mentioned that both products are based on the same input data and are spatially related to each other by means of the MSR.

## 4.5 Assessment of High Resolution VTEC Models

Since the product 'othg' outperforms all other products used in the previous sections we now compare it with UPC's product
5  'uqrg' (Roma-Dollase et al., 2017) which provides smaller values in the relative standard deviation of their performed dSTEC analysis in comparison to the products of other IAACs. 'uqrg' is a rapid product and provided with a sampling interval $\Delta T = 15$ minutes. Since the station 'NKLG' is not used in the calculation of 'uqrg' it is excluded from the calculation of the overall RMS value shown in the legend. As can be seen, the RMS values vary between 0.5 and 1.8 TECU but are mostly below 1.0 TECU. The dominant RMS value of 'uqrg' at the station 'CHPI' reduces its quality significantly. In the case of neglecting
10  'CHPI', the mean RMS value of 'uqrg' decrease to 0.59 TECU. Summarizing these investigations, we can state that the overall quality of the two products is very close to each other. Considering the fact that 'othg' is a NRT product with a latency of less



**Table 4.** Results in percentage of the comparisons of the high resolution products 'ophg' with with the low resolution products 'oplg'. Positive (green and red colored) numbers mean an improvement, negative (blue colored) values a reduction of the quality.

|      | o2lg    | o1lg    | otlg    |
|------|---------|---------|---------|
| o2hg | 2.2 %   | -12.5 % | -16.9 % |
| o1hg | 21.7 %  | 10.0 %  | 6.5 %   |
| othg | 26.1 %  | 15.0 %  | 11.7 %  |

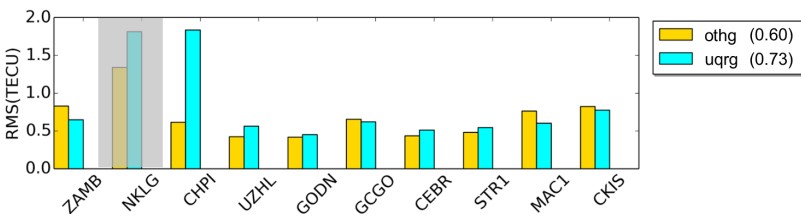

**Figure 14.** RMS values for the products 'uqrg' and 'othg' computed at 8 IGS receiver stations during September 2017. The values in the parentheses in the legend are the average RMS values over all 8 receiver stations for the entire test period between September 1 and September 30, 2017.

than 3 hours, it also outperforms 'uqrg' which is a rapid product with a latency of around 1 day; cf. Table 2. However, for a final assessment further validation studies have to be performed between the different products.

## 5 Conclusions and Outlook

This paper presents an approach to model VTEC from NRT GNSS observations only by generating a MSR based on B-splines; the unknown model parameters are estimated by means of an KF. Based on this approach, a number of products have been created which differ both in their spectral content and in their temporal resolution. From our investigations we state that the MSR provides B-spline models comparable to the standard GIMs of the IAACs, mostly based on SH expansions up to degree $n_{max} = 15$. As the core of the numerical study we compare our results with the most prominent VTEC maps of the IAACs to rate the quality. Since the dSTEC analysis is the most frequently used validation method, we abandon here a comparison with satellite altimetry products. As the outcome of the validation studies, it can be stated that the high resolution product 'othg' outperforms all products used within the selected time span of investigation.

Besides the facts, that our models can handle data gaps because of the utilization of localizing basis functions, the application of a KF to include a dynamic prediction procedure and the use of the MSR to create products of different spectral content at the same time, it shall be mentioned that DGFI-TUM's products . . .



... are based on NRT GNSS observations only, i.e. are using input data with a latency of less then 3 hours (in opposite 'codg' relies on post-processed data with a latency of larger than 3 week and 'uqrg' on rapid data with a latency of at least 1 day); cf. Table 2

... rely on specially developed software modules, cf. Fig. 15, e.g. the pre-processing module using ultra-rapid orbits

... can be disseminated to users with a delay of two to three hours.

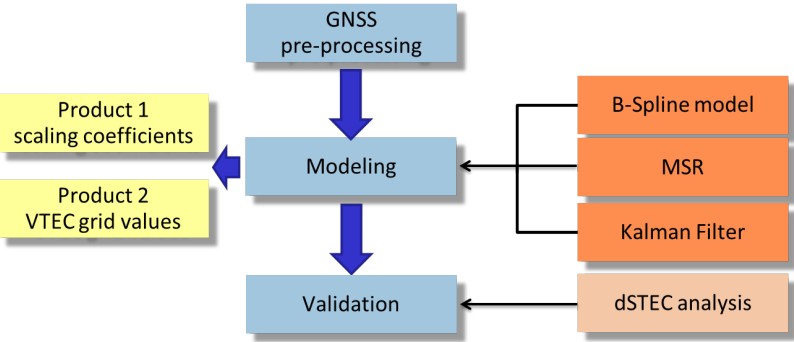

**Figure 15.** DGFI-TUM's processing modules, including (blue colored boxes) the download and pre-processing module for GNSS observations, the modeling module by means of B-splines, MSR and Kalman filtering (orange colored boxes) with possible output as Product 1 and Product 2 (yellow colored boxes) and the validation module.

In general, the dissemination of these products to users can happen in two different ways, namely based on estimated scaling coefficients (Product 1) or by calculated VTEC grid values (Product 2). For RT applications, however, the dissemination in terms of Product 1 is preferred, in particular the usage of the RTCM format. In the scope of the developments in the recent years, RT applications become more prominent, e.g. in unmanned or autonomous vehicle development, and thus, the restriction

of the RTCM message to allow only for SH coefficients needs urgently to be discussed. Especially from the point of view that there are also other modelling methods, a modification of the RTCM format would be appropriate. The MSR allows for a significant data compression obtained due the step-wise downsampling of the scaling coefficients according to the pyramid algorithm. Details represented in the signal $f_{J_1, J_2}$ of the $0^{th}$ step are stored in wavelet coefficients for the following steps, cf. Fig. 5. A large number of estimated wavelet coefficients are characterized by absolute values smaller than a given threshold

and thus, most of them can be neglected for the reconstruction of the original signal. Hence, the overall number of scaling and wavelet coefficients can be reduced drastically. Considering this powerful feature of data compression we propose to replace the scaling coefficients of the highest levels by the significant wavelet coefficients of the lower levels for a definition of an alternative and more appropriate format for data dissemination in terms of Product 1.

The presented results encourage the further development of high accuracy VTEC maps. By extending the models by a fourth

dimension, i.e. modelling of the electron density directly, inaccuracies due to the mapping function can be avoided. Since To model the vertical structure of the electron density has to be modelled, additional observations have to be incorporated,



e.g. from DORIS, satellite altimetry and ionospheric radio occultations. This would mitigate the inhomogeneity of the data distribution and thus even higher levels of the B-spline expansion can be chosen.

*Acknowledgements.* We are grateful to the Bundeswehr GeoInformation Centre (BGIC) and the German Space Situational Awareness Center (GSSAC) for funding the project "Operational tool for ionosphere mapping and prediction" (OPTIMAP). The presented approach was

5   developed within this framework.

The authors express their thanks to services and institutions for providing the input data: IGS and its data centers, the Center for Orbit Determination in Europe (CODE, Berne, Switzerland) and the Universitat Politècnica de Catalunya/IonSAT (UPC, Barcelona, Spain). Furthermore the authors acknowledge the developers of the Generic Mapping Tool (GMT) mainly used for generating the figures in this work.

10   This work was supported by the Technical University of Munich (TUM) in the framework of the Open Access Publishing Program.





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
