# Peer review of "High Resolution Vertical Total Electron Content Maps Based on Multi-Scale B-spline Representations"

_Annales Geophysicae, 2019_

## Referee Comment (RC1) · Ilya Edemskiy (Referee) · 14 May 2019

The paper presents an interesting investigation, containing a good review of current state in global ionosphere maps production and a presentation of a promising technique for calculation of near-real time maps of high spatial resolution. The presented method to be used for RT applications and it would be better to describe in more details how much information should be transmitted to user and if its amount depends on space weather conditions.

I would also ask the authors to correct or clarify some parts of the article (P – page, L – lines):

P5L6: What kind of difference between Europe and Oceania is possible to see in fig.

[Figure]

9Left? Both regions demonstrates "smooth" VTEC.

P6, footnote: where is this reference in the text?

P7L18: DGFI-TUM abbreviation is not introduced properly

P7L9: Term "sampling interval" for latitude and longitude ($\Delta\lambda$, $\Delta\varphi$) is not really clear. Typically the term means an interval between measurements, but the authors use it for a model explanation. The term should be explained more carefully, especially since the used notation can be easily misunderstood as one for the resolution intervals ($\Delta\Lambda$, $\Delta\Phi$).

fig4. Notations $\Delta\Lambda$ and $\Delta\Phi$ are located badly since it looks like that q = $\Delta\Phi$ and p = $\Delta\Lambda$

P13L15-16: ". . . quality . . . decreases with the spatial resolution intervals". Resolution intervals decrease or increase?

P17L17-18: Notation Is is not neither introduced nor explained.

P18L28 – P19L15. The sentence is not divided in a good way. Please reformulate.

P22L9: Linear interpolation gives no information about detailed structure of ionosphere and, I would say, only makes an illusion of higher resolution. I recommend to present real grid figures to show real quality of the model.

P22L13-15: The statement is not fully correct since e.g. Southern Pacific Ocean also does not contain GNSS station, but demonstrates the same standard deviation level as the region of Northern America.

P23L3: Are the indices 4, 3 over $\sigma$ correct? Should not they be "5, 3"?

P23L4, 8: Both maps have the same resolution intervals, but different "sampling intervals". Presentation of the "sampling intervals" here is more necessary, when resolution ones could be described outside of this MS products block.

P23L9-15. I recommend to use "symbol" instead of "digit" (here and further) since usually "digit" means a number. All the presented description is too complicated. I recommend you to explain it this way: "'oABg' where A is a temporal resolution (10 min('t'), 1 hour('1') and 2 hours ('2') and B is high ('h') or low ('l') resolution." Letters 'o" and 'g' are the same for all your abbreviation. It is also not necessary to repeat the explanation in P25L7-11.

P24L16-17 and fig 10: Specify the independent stations names and show them by color in figure. It is not clear for me why do not you use only independent stations for validation. Why do not you take other 10 stations located close to the ones you choose?

P25L2: If some threshold was used, specify its value in the text.

P25L5-6. It would be much more representative to show total number and a number of the used coefficients.

P30L21: Phrase "Since To model. . ." is not clear.

---

## Referee Comment (RC2) · Anonymous Referee #2 · 29 May 2019

This work presents a very detailed analysis for obtaining VTEC maps of high resolution. Although the reading is a little bit obscured by so much mathematics, I think this work is of much interest for the community using and needing this type of interpolation and procedures. So, in my opinion it is acceptable for publication in this Journal in its present form. I have only two very minor comments, which the author can take into account, or not. Page 2, line 13: Even though "bending" can be neglected in some aplication I think it is the principle of operaton of many devices, as radars for example for which the ability of the ionosphere to bend and finally reflect HF signals is extremely important. Page 2, line 17: Higher order effects are also affected by the Earth's magnetic field.

---

## Author Comment (AC1) · 29 May 2019

Dear Dr. Edemskiy,
Thank you very much for your comments, questions and suggestions. We added most of you suggestions to the manuscript and modified the parts which were not clearly described. The modified manuscript will be uploaded as soon as we could incorporate the comments of the second referee.

We respond to your questions indicated with P – page and L – line related to the position in the manuscript.

Additionally, we add the corrected standard deviation map of Fig (8) and our comments as .pdf version in the attachment.

[Figure]

Best regards
Andreas Goss

**P5L6: What kind of difference between Europe and Oceania is possible to see in fig. 9Left? Both regions demonstrates "smooth" VTEC.**
Figure 1 shows the global distribution of IPPs from GPS and GLONASS. Due to the distribution of stations, the observation points (in this case IPPs) are mostly located over continents. While over Europe and North-America dense clusters of observations are available, in other continental regions such as Africa or Oceania the data distribution is rather sparse. Usually the signal structure over Europe is rather smooth while in equatorial regions such as Oceania stronger variations in VTEC especially at local noon time exist. Figure 9 shows the VTEC map of September 6, 2017 at 13:00 UT. Both regions, Europe and Oceania show smooth VTEC structures. In order to represent stronger variations in VTEC and thus finer signal structures during local noon time dense data coverage is necessary, which is, however, generally not given in equatorial regions. Consequently, there is a strong incongruity between the data distribution and the signal structure.

**P6: footnote: where is this reference in the text?**
We placed the reference to the footnote to the correct position in the text. Thank you for noticing this error.

**P7L9: Term "sampling interval" for latitude and longitude ($\Delta\varphi$, $\Delta\lambda$ ) is not really clear. Typically the term means an interval between measurements, but the authors use it for a model explanation. The term should be explained more carefully, especially since the used notation can be easily misunderstood as one for the resolution intervals ($\Delta\Phi$,$\Delta\Lambda$).**
In P7L9 the values $\Delta\phi$ and $\Delta\lambda$ are used as the sampling intervals of the input observations, here seen as global mean sampling intervals related to latitude and longitude. According to the sampling theorem on a sphere, $\Delta\varphi$ and $\Delta\lambda$ are necessary to calculate the maximum degree of a SH expansion. In Section 2.2.1 (P9L14) and in Section 2.2.2 (P11L6) the resolution levels $J_1$ and $J_2$ for the B-spline expansion are defined in dependence of the values $\Delta\varphi$ and $\Delta\lambda$. In Section 2.3 (P11L13) the values $\Delta\varphi$ and $\Delta\lambda$ are used again as the sampling intervals in order to derive the spectral relation between the two model approaches (SHs or B-splines). In Section 2.4 we distinguish between the input sampling intervals $\Delta\varphi$ and $\Delta\lambda$ of the observations and the output resolution intervals $\Delta\Phi$ and $\Delta\Lambda$ of the VTEC model. Whereas $\Delta\Phi$ and $\Delta\Lambda$ are defined by the observations, the output resolution intervals and must be chosen in dependence of the spectral content of the output signal, i.e. the VTEC model. **fig4. Notations and are located badly since it looks like that q = $\Delta\Phi$ and p = $\Delta\Lambda$ :** The figure is now modified. Thank you for that comment.

**P7L18: DGFI-TUM abbreviation is not introduced properly:**
The abbreviation DGFI-TUM is now introduced within the abstract.

**P13L15-16: "... quality ... decreases with the spatial resolution intervals". Resolution intervals decrease or increase?**
The quality of the interpolated VTEC value decreases with increasing resolution intervals. We corrected the statement in the new manuscript.

**P17L17-18: Notation is not neither introduced nor explained.**
The in the size definition for the vectors of observations and measurement errors was introduced in Section 2 in Eq. (8) as the total number of given observations at the time moment. We added a reference in the new manuscript.

**P18L28 – P19L15: The sentence is not divided in a good way. Please reformulate.**
The sentence is now separated into two parts. You were right, it was quiet long and not understandable.

**P22L9: Linear interpolation gives no information about detailed structure of**

**ionosphere and, I would say, only makes an illusion of higher resolution. I recommend to present real grid figures to show real quality of the model.**

The maps shown in Fig. 8 are calculated on a much denser grid as it was described in P22L9-10 with " . . . we compute the function values on a much denser grid by using the interpolation formula (34)".

**P22L13-15: The statement is not fully correct since e.g. Southern Pacific Ocean also does not contain GNSS station, but demonstrates the same standard deviation level as the region of Northern America.**

That is true, thank you very much for the statement. The standard deviations have been calculated newly for the example Figures 8 and 9 and updated in the new manuscript. The standard deviation maps in the first manuscript were incorrect. Thank you for the comment. The new standard deviation map is attached in Fig. 1 to this responses.

**P23L3: Are the indices 4, 3 over $\sigma$ correct? Should not they be "5, 3"?**

That's totally correct. We changed the numbers in the equation.

**P23L8: Both maps have the same resolution intervals, but different "sampling intervals". Presentation of the "sampling intervals" here is more necessary, when resolution ones could be described outside of this MS products block.**

The two Multi-Scale Products 2 are here defined as VTEC grids with output resolution of and as it was described in Section 3.3 (P19L8-14). The scaling coefficients of level and for the product 'ophg' are estimated using data given with approximated sampling intervals of and in the Sun-fixed coordinate system as introduced in the context of Eq. (64) on P21L21. For generating the product 'oplg' the 1-D pyramid algorithm shown in Fig. 6 was applied. Since this is the application of the processing chain for the two Multi-Scale Products 2 it not necessary to provide additional information about the sampling intervals. We added the resolution intervals and for the two products because these are standard values for the IGS products.

**P23L9-15: I recommend to use "symbol" instead of "digit" (here and further)**

**since usually "digit" means a number. All the presented description is too complicated. I recommend you to explain it this way: "'oABg' where A is a temporal resolution 10 min('t'), 1 hour('1') and 2 hours ('2') and B is high ('h') or low ('l') resolution." Letters 'o" and 'g' are the same for all your abbreviation. It is also not necessary to repeat the explanation in P25L7-11.**

It is totally fine to use the phrase "symbol" since it would fit to both numbers and letters. In our opinion, the usage of additional letters to describe the symbols makes the description more complicated. We would like to stay by the usage of 'p' for the temporal output sampling symbol as well as 'l' and 'h' for the spectral resolution, since the latter already indicate the meaning of "low" and "high". We removed the additional description of the symbols at P25L7-11, as you suggested.

**P24L16-17: Specify the independent stations names and show them by color in figure. It is not clear for me why do not you use only independent stations for validation. Why do not you take other 10 stations located close to the ones you choose?**

It is an up to date discussion which stations should be used for the dSTEC analysis. There are three possibilities to compute a fair comparison:

1. Usage of stations which are independent from all models to be compared
2. Usage of stations which are all used for the computation of the models
3. A mixture of 1. and 2.

Within the paper we chose the third option, because we wanted to include all (NRT) stations of the IGS network to compute VTEC maps with high quality. Since the dSTEC analysis should be based on stations with a global homogeneous distribution, we used also stations for the comparison which are located in isolated regions and which are required for the model computations.

**P25L2: If some threshold was used, specify its value in the text.**

P25L5-6: It would be much more representative to show total number and a number ofthe used coefficients.

The paragraph only gives information about the advantages of MSR for data compression. It means that by usage of the pyramid algorithm the number of significant coefficients can be reduced in order to disseminate data with smaller size in terms of storage problems. There was no threshold used and the data compression was not investigated. A study about this advantageous feature of the MSR is no content of this paper and will be applied and published in the future

**P30L21: Phrase "Since To model. . ." is not clear:**
The phrase is updated, thank you for that comment.

Please also note the supplement to this comment:
https://www.ann-geophys-discuss.net/angeo-2019-32/angeo-2019-32-AC1-supplement.pdf
* * *
[Figure]

**Fig. 1.**

---

## Author Comment (AC2) · 25 Jun 2019

Dear Referee,
Thank you very much for your comments.

**Page 2, line 13:**
That is right. Also for radio occultations the bending angle of the signal is used to derive the electron density within the ionosphere.
At this stage of the paper, we tend to derive the equation for the ionospheric delay due to the electron density, therefore we would stay with the statement and not add to much details about the bending to the manuscript.

[Figure]
* * *
**Interactive**
**comment**

**Page 2, line 17:**
The higher order ionospheric effects are also caused by the magnetic field, as you mentioned it. We will add this to the manuscript in the list of possible causes for higher order effects. Thank you.

Best regards
Andreas Goss